# Integrated computational and experimental immunoengineering of adeno-associated virus capsid T cell epitopes in mice

Sojin Bing[1], Arya Eskandarian[2], Sean Smith[2], Abdul Mohin Sajib[1,4], Stephanee Warrington[1], Sima Saleh[1,5], Susana S Najera [1], Rebecca J. D'Esposito[3], Luis Santana-Quintero [2] & Ronit Mazor [1] ✉

Adeno-associated virus (AAV) vectors are widely used in gene therapy, but their immunogenicity remains a significant challenge, limiting long-term efficacy and the feasibility of repeated administration. In this study, we combine computational prediction with experimental validation to engineer AAV9 capsids with reduced immunogenicity. To facilitate this, we developed the Epitope Modification and MHC Prediction (EMMP) pipeline, which systematically automates the evaluation of amino acid substitutions for their predicted effects on major histocompatibility complex (MHC) presentability. Using this pipeline, we modify a CD4+ T-cell epitope in the AAV9 capsid that is identified and characterized as a proof-of-concept. Two mutant variants, R312H and R312Q, are selected and evaluated for transduction efficiency in vitro and immune response modulation in vivo. Notably, R312Q shows a significant reduction in T-cell activation and anti-AAV9 antibody production, albeit with a slight reduction in transduction at low multiplicities of infection (MOI). These results demonstrate a rational approach for optimizing AAV vector design, with potential applications for improving the safety and efficacy of gene therapy.

Adeno-associated virus (AAV) vectors have emerged as a leading platform for gene therapy due to their ability to mediate long-term gene expression with minimal toxicity. Several AAV-based gene therapies have received regulatory approval, including treatments for inherited retinal diseases and spinal muscular atrophy[1,2]. However, a major challenge in AAV-mediated gene therapy is the host immune response against the viral capsid, which can lead to vector clearance, loss of transgene expression, and limitations in repeat dosing[3].

The immune response to AAV involves both humoral and cellular components. Pre-existing antibodies against AAV, commonly present due to natural infections, can prevent successful transduction[4]. Additionally, T-cell responses directed against AAV capsid proteins contribute to immune-mediated clearance of transduced cells[5]. CD4+ T cells play a critical role by facilitating the production of anti-AAV antibodies and promoting CD8+ T-cell responses, which have been implicated in the loss of transgene expression observed in clinical trials[6,7].

To mitigate AAV immunogenicity, various strategies have been explored, including capsid engineering to evade neutralizing antibodies[8], transient immunosuppression[3], B cell depletion[9], and cleavage of human IgG[10]. However, little attention has been given to the targeted modification of T-cell epitopes within the capsid. The major histocompatibility complex (MHC) plays a crucial role in adaptive immunity by binding antigenic peptides and presenting them to

[1]Office of Gene Therapy, Office of Therapeutic Products, Center for Biologics Evaluation and Research, U. S. Food and Drug Administration, Silver Spring, MD, USA. [2]Office of Biostatistics and Pharmacovigilance, Center for Biologics Evaluation and Research, U. S. Food and Drug Administration, Silver Spring, MD, USA. [3]Waters Corporation, Milford, MA, USA. [4]Present address: Pfizer Inc., Andover, MA, USA. [5]Present address: Division of Cardiology, Department of Medicine, Johns Hopkins University School of Medicine, Baltimore, MD, USA. ✉e-mail: ronit.mazor@fda.hhs.gov

T cells. Even minor alterations in peptide-MHC interactions can significantly impact T cell activation[11].

Previously, we demonstrated that rational epitope modification through chimeric design could successfully eliminate immunodominant T-cell responses to AAV9 in human PBMCs[12]. In that study, multiple amino acids within the immunodominant epitope were replaced with corresponding sequences from AAV5, resulting in the complete elimination of T-cell activation. However, the chimeric approach was limited to cases where suitable non-immunogenic sequences existed in related viral serotypes. Building upon this foundation, the current study addresses the need for a more systematic and broadly applicable approach to epitope modification. Here, we develop the Epitope Modification and MHC Prediction (EMMP) pipeline to comprehensively evaluate all possible single-point mutations, insertions, and deletions within immunogenic epitopes, enabling rational design even when suitable chimeric sequences are not available. Furthermore, we validate this approach in vivo using a mouse model and demonstrate its broader applicability to other viral vectors.

In this proof-of-concept study using a mouse model, we seek to identify immunogenic T-cell epitopes within the AAV9 capsid. Using an overlapping peptide library spanning the AAV9 VP1 protein and in vivo immunization studies in BALB/c mice, we identify a CD4 T cell epitope centered around residue R312. This epitope serves as a rational target for further engineering aimed at immune evasion.

However, rationally modifying T-cell epitopes requires a careful balance. While the goal is to reduce MHC binding affinity and prevent T-cell recognition, any introduced mutations must also preserve the structural integrity and biological function of the protein. In the context of AAV capsid engineering, this is particularly important, as even single-point mutations can affect capsid assembly, genome packaging, cell binding, or transduction efficiency[13]. Moreover, the computational burden of evaluating every possible amino acid substitution within an epitope across multiple MHC alleles is impractical due to time constraints and the potential for errors.

To address these challenges, we develop the EMMP pipeline, a computational framework that leverages the Immune Epitope Database (IEDB) and systematically automates the application of existing MHC-binding prediction algorithms to evaluate MHC binding affinity for all single-residue substitutions within a given epitope. EMMP enables high-throughput analysis across multiple alleles, identifies potential mutations that may reduce MHC presentability, and provides visualization of results to facilitate interpretation. This pipeline significantly accelerates the design of deimmunized protein variants and is publicly available via the FDA/GIT repository.

We apply EMMP to the CD4⁺ T-cell epitope identified in AAV9, using its predictive capabilities to screen for mutations likely to reduce binding to MHC class II molecules. Based on these predictions, we select candidate mutations and experimentally evaluate their impact, demonstrating the effectiveness of EMMP in streamlining the identification of optimal deimmunization strategies. Specifically, we engineer two AAV9 variants (R312H and R312Q) to disrupt HLA class II presentation and consequent T-cell recognition. We assess the effects of these mutations on transduction efficiency, biodistribution, and immune responses. Our findings demonstrate that the R312Q variant effectively eliminates T-cell activation and reduces anti-AAV antibody production while maintaining vector biodistribution, providing a potential strategy for improving AAV-based gene therapies.

## Results

### Identification and characterization of T-cell epitopes in AAV9 in BALB/c mice

To identify the T cell epitopes in the AAV9 capsid, BALB/c mice were immunized with AAV9 in the presence of CpG adjuvant. Splenocytes from 18 mice were isolated and stimulated with 31 peptide pools spanning the AAV9 capsid sequence in three experimental batches. IL-2 ELISpot assays revealed strong responses in pools 7, 21, and 22, which were significantly higher than the no-peptide control (Fig. 1A). Notably, peptides 102 and 103 were commonly present in these pools (Fig. 1B). Further analysis of individual peptides confirmed that peptides 102 and 103 elicited IL-2 and IFN-γ production in a dose-dependent manner (Fig. 2A–C), indicating that they contain immunodominant epitopes corresponding to amino acids 304–318 and 307–321 of the AAV9 capsid.

To determine the responding T-cell subset, splenocytes from immunized mice were separated into CD4⁺ and CD8⁺ T-cell populations before peptide restimulation. Only CD4⁺ T cells responded to the peptides, while CD8⁺ T cells showed no response in ELISpot assays (Fig. 2D, E), suggesting that the identified epitope is recognized by CD4⁺ T cells.

### Predicting mutations in a murine AAV9 CD4⁺ T cell epitope using EMMP

AAV9 epitopes previously mapped in human samples identified a region spanning peptides 103–105 that elicited responses in 23% of donors[12]. While this human epitope is in close proximity to the mouse epitope identified here, the core regions differ. Additionally, we analyzed the MHC class II binding affinity of this epitope using the NetMHCII2.3 method from the IEDB database, which was the most up-to-date version available at the time of analysis. The sequence NNWGFRPKRLNFKLF was predicted to have the strongest binding affinity, ranking in the 1.2 percentile, confirming that peptides 102 and 103 are likely processed and presented by MHC class II molecules (H2-IEd) in BALB/c mice (Supplementary Table 1).

We developed three pipelines that utilize MHC binding affinities predictions, one for each possible mutation: a replacement, an insertion, and a deletion. These codes are based on the concept that minor modifications to the peptide sequence can significantly impact its binding affinity to an MHC molecule. The EMMP automatically generates MHC presentability predictions for both MHC-I and MHC-II molecules using existing prediction algorithms for all three cases and stores the results for each algorithm in different folders that facilitate the location of these results. A flowchart illustrating the EMMP workflow is shown in Fig. 3A. For this study, EMMP utilized NetMHCII2.3 for MHC class II binding predictions, which was the most current and IEDB-recommended method available at the time of analysis. However, we acknowledge a critical limitation: different prediction algorithms can yield substantially conflicting results. As shown in Supplementary Table 2, while NetMHCII2.3 predicted our selected mutations (R312H and R312Q) as having significantly reduced MHC presentability, the more recently recommended NetMHCIIPan4.0 EL algorithm predicts these same mutations would retain strong MHC presentability similar to wild type. This discrepancy means that using newer algorithms, these mutations would not have been selected for experimental testing, highlighting how algorithm choice fundamentally impacts candidate selection. The current version of EMMP has been updated to use NetMHCIIPan4.1, but this comparison underscores that EMMP's primary value lies in systematic automation rather than enhanced prediction accuracy[14]. EMMP can be used to analyze binding prediction of a single MHC molecule, provided epitope restriction has been established, or for promiscuity analysis by selecting multiple MHCs.

We subsequently applied EMMP analysis to identify potential mutations and visualize the results as a heatmap (Fig. 3B). In the case of replacement, single-point mutations at aa 308 (F), aa 309 (R), and aa 312 (R) in the capsid were predicted to significantly reduce MHC presentability, suggesting that these residues play a crucial role in MHC class II binding. For insertion mutations, introducing D or G between aa 308 (F) and aa 309 (R) or between aa 309 (R) and aa 310 (P) was predicted to decrease MHC presentability. Similarly, deletion of aa 308 (F) and aa 309 (R) resulted in a marked reduction in MHC

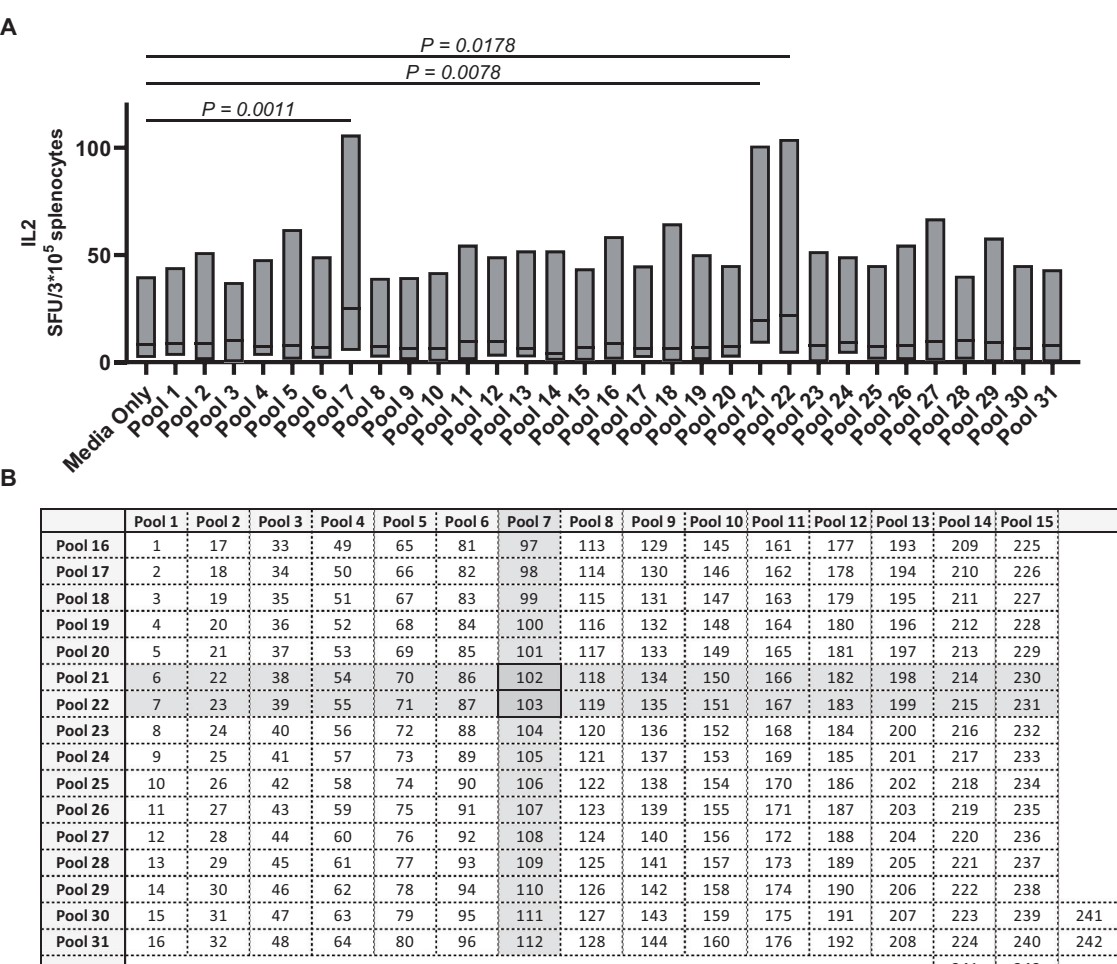

**Fig. 1 | T-cell epitope mapping of AAV9 peptides in BALB/c.** Female mice ($n = 18$) were injected with 1E11 vg of AAV9 and 5 μM of CpG intraperitoneally. On days 9–10, mice were euthanized, and splenocytes were isolated from each mouse separately. Splenocytes from each individual mouse were plated in IL-2 ELISpot-coated plates and stimulated with peptide pools. **A** ELISpot response to peptide pools spanning the sequence of the AAV9 capsid. $n = 18$ biological replicates (independent mice). Data were pooled from three independent experimental batches. Each mouse was processed independently, and ELISpot assays were performed in triplicate for each individual mouse. Values are presented as mean ± SEM. In the box plot, the center line represents the median, the bounds of the box represent the 25th and 75th percentiles, and the whiskers indicate the minimum and maximum values. The y-axis represents the number of spot-forming units (SFU) per 3E5 cells. Significance was calculated using a one-way ANOVA with Dunnett's multiple comparisons test. **B** Matrix pools are used for T-cell epitope mapping of the AAV9 capsid. Source data are provided as a Source data file.

presentability. Based on these predictions, we selected a total of 26 candidate mutant sequences for further validation, comprising 20 replacement mutations, 4 insertion mutations, and 2 deletion mutations (Table 1). Importantly, these predictions were generated using NetMHCII2.3. When the same sequences were analyzed using more recent algorithms (NetMHCIIPan4.0 EL), substantially different results were obtained (Supplementary Table 2), demonstrating the algorithm-dependent nature of computational predictions and reinforcing the necessity of experimental validation for any computationally selected candidates.

### Semi-high-throughput screening of AAV9 mutants for transduction efficiency

To select the mutants among those 26 candidates that we previously identified, we produced AAV particles and screened their transduction efficiency (Fig. 4). To enable a simultaneous and side-by-side screening of all mutants, we utilized Viral Production Cells 2.0, a suspension-adapted HEK293 cell line with high cell density and high yield[15]. Triple transfection was performed in 12-well plates, where each plate contained wild-type (WT) control, sham control, and a maximum of 10

mutant variants. Following transfection, the supernatants from three 12-well plates were collected and subjected to PEGylation in a 96-deep-well plate to concentrate AAV particles. The purified AAV particles were then directly applied to HeLa cells seeded in 96-well plates without quantification. This approach significantly reduces time and cost by allowing rapid selection of AAV lead candidates within just 7 days from plasmids to final candidate, compared to traditional methods that often require weeks due to labor-intensive production and purification steps (Fig. 4A).

Transduction efficiency of the 26 candidate mutants was assessed using a *GFP* transgene. Transduced HeLa cells in the 96-well plate were imaged directly using a fluorescent ELISpot reader supporting rapid quantification of transduced cells with no need for cell detachment (Supplementary Fig. 1). Subsequently, the percentage of GFP-positive cells was further quantified via flow cytometry (Fig. 4B) for higher sensitivity comparison of the highly positive wells. Among the 26 candidates, R312H and R312Q exhibited transduction efficiency comparable to the WT control and the previously reported AAV9_VI mutant[12]. R312E also facilitated transduction; however, its efficiency was lower than that of the other two mutants. It is important to note

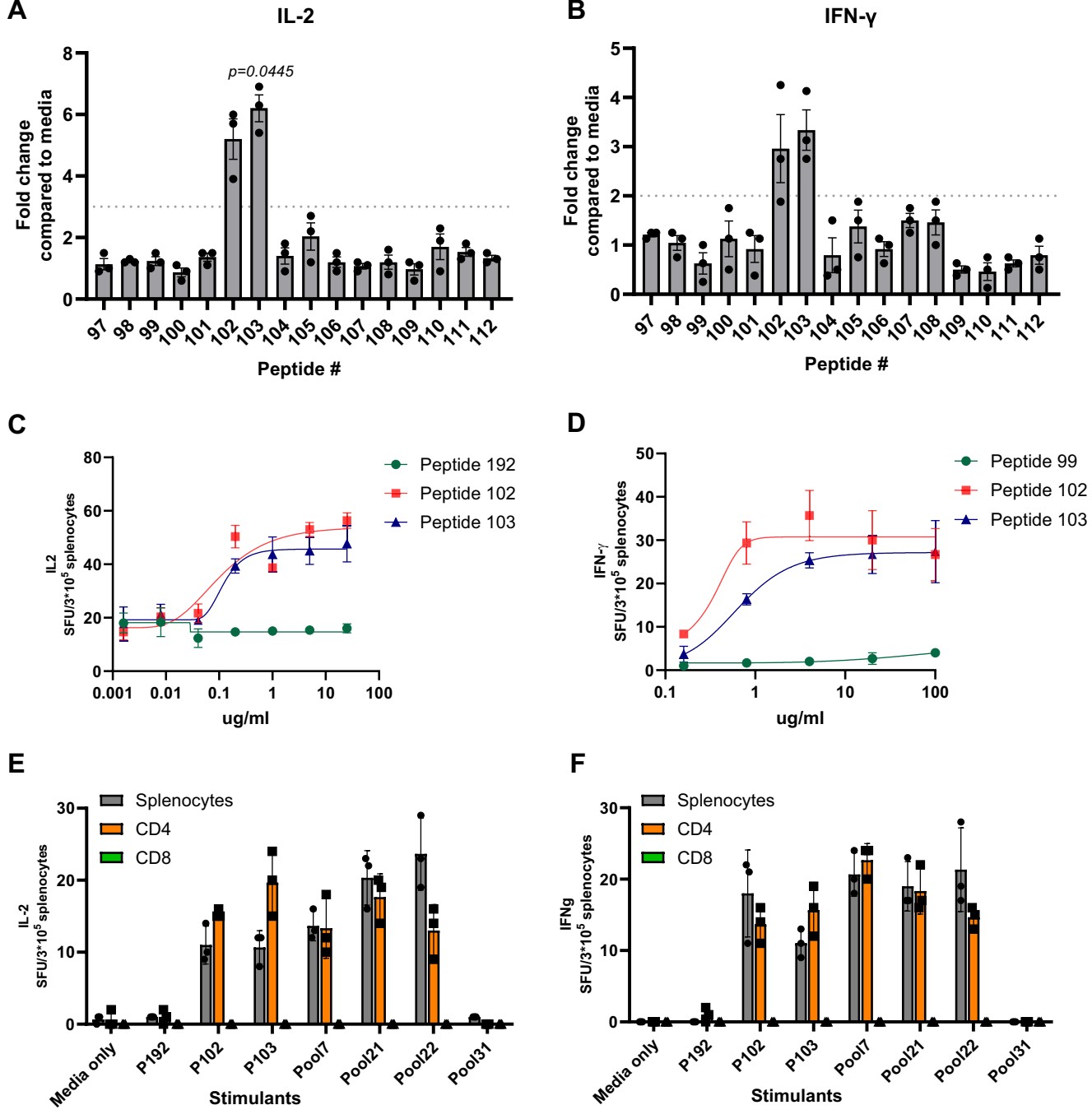

**Fig. 2 | Determination and characterization of AAV9 T cell epitope.** Female mice were injected with 1E11 vg of AAV9 and 5 μM of CpG intraperitoneally. Splenocytes were stimulated with the appropriate peptides for a day in ELISpot plates, IL-2 (**A**) and IFN-γ (**B**) responses to the peptides that comprise pool 7. The dotted line represents two- or three-times background. Data are presented as mean ± SEM from three independent biological replicates. Statistical significance was determined using a non-parametric one-way ANOVA (Friedman test) followed by post-hoc multiple comparisons with Dunn's test. IL-2 (**C**) or IFN-γ (**D**) responses to different doses of peptides. The titration curves for peptides 99, 102, and 103 are represented by green circles, red squares, and blue triangles, respectively. The value $n = 3$ indicates the number of independent biological samples (e.g.,

splenocytes from individual mice). Data are presented as mean values ± SEM. CD4 or CD8 T cells were isolated from pooled splenocytes of AAV9-injected mice ($n = 3$) and stimulated with the indicated peptides or pools to assess IL-2 (**E**) or IFN-γ (**F**) production. Gray bars represent total splenocytes, orange bars represent isolated CD4+ T cells, and green bars represent isolated CD8+ T cells. Values are presented as mean ± SD. While statistical significance was not achieved due to small sample size ($n = 3$) and zero variance in the CD8+ group, the biological difference is evident with CD8+ T cells showing no detectable response (all values = 0) compared to robust responses in CD4+ T cells. Source data are provided as a Source data file. SFU spot-forming unit.

that this screening was conducted without precise determination of the multiplicity of infection (MOI) and was intended solely for preliminary selection. Based on this screening, the 26 initial candidates were narrowed down to two promising mutants.

**Transduction efficiency of WT AAV9 and mutated AAV vectors in cell lines**

The two candidates, R312H and R312Q, were generated and tested for their impact on vector production and transduction efficiency. All

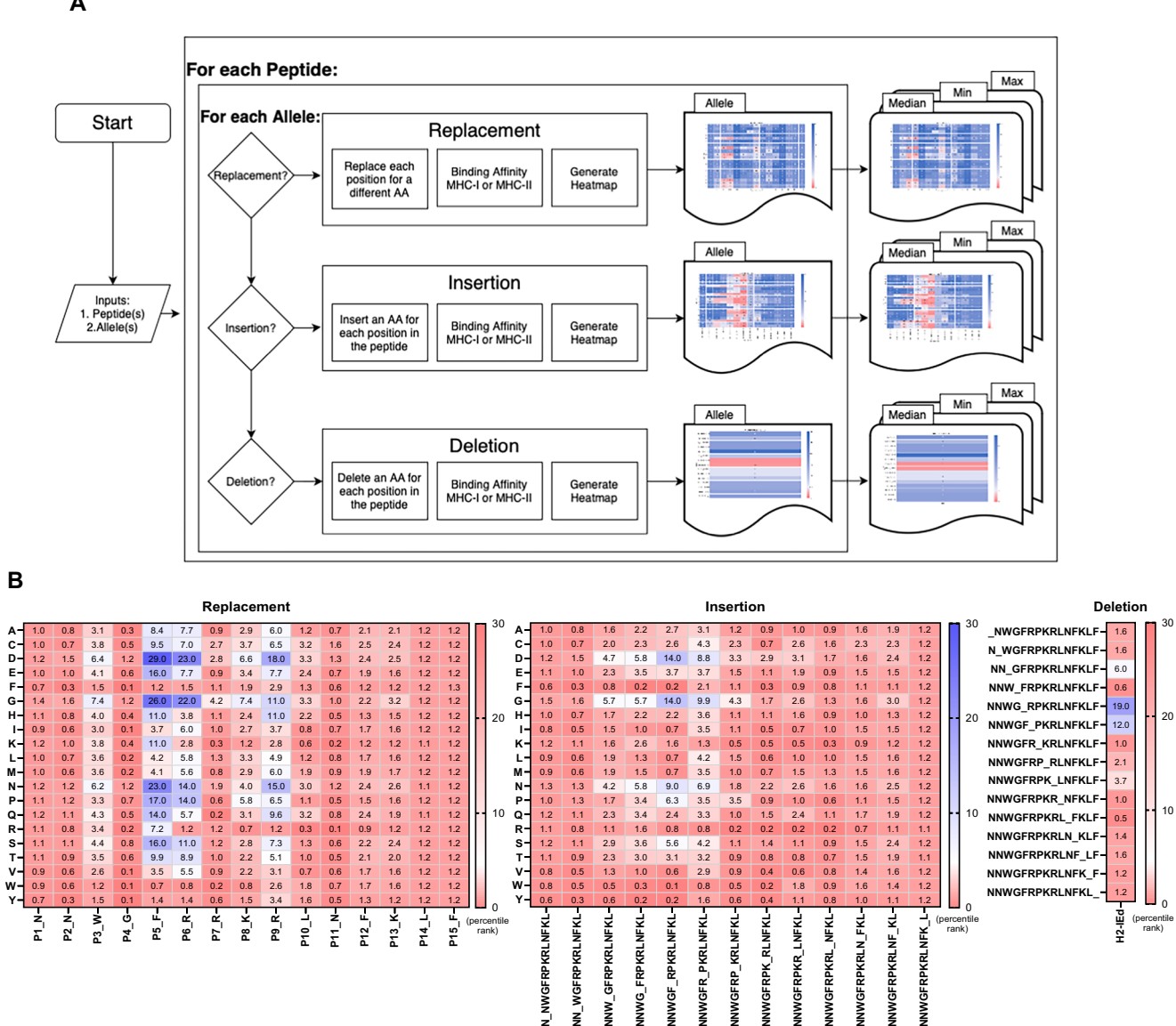

**Fig. 3 | Flowchart and representative output of Epitope Mutator & MHC Binding Predictor (EMMP). A** Flowchart of EMMP. For each input peptide and allele combination, a novel peptide sequence is iteratively generated through a series of modifications, including single amino acid (AA) substitutions, insertions, and deletions. Specifically, all possible AA replacements, insertions at every position, and single-residue deletions at each site along the peptide sequence are systematically evaluated for their impact on MHC presentability. **B** Exemplary output from the deimmunization algorithm for peptide sequence NNWGFRPKRLNFKLF, suggesting modifications to minimize immune recognition. The prediction results are color-coded based on their impact on the predicted binding affinity. Mutations with negligible or minimal effects on the prediction value are displayed in white. In contrast, mutations that increase the predicted MHC presentability (i.e., unfavorable mutations) are highlighted in red, whereas those that decrease the MHC presentability (i.e., favorable mutations) are shown in blue.

AAV9 variants were successfully produced via triple transfection in HEK293 cells, with no significant differences in viral yield, genome-to-particle ratio, percentage of empty and partial capsid, or protein purity (Supplementary Fig. 2).

The transduction efficiencies of WT and mutant AAV9 vectors were evaluated in HEK293T and HeLa cells across a range of multiplicities of infection (MOI). Flow cytometry analysis showed that WT and R312H transduced cells with comparable efficiency, while R312Q exhibited a reduction of 4.6 and 14.5% in transduction efficiency in HeLa and HEK293T, respectively, particularly at low MOI (Fig. 5A, B and Supplementary Fig. 3A, B). There were no significant differences at high MOIs, but the area under the curve (AUC), which includes all MOI data, showed significance between WT and the R312Q variant. Similar results were observed using an alternative transgene (NanoLuc),

detectable with higher sensitivity (Supplementary Fig. 3C–F). These findings suggest that R312H has minimal impact on in vitro transduction, whereas R312Q has reduced transduction efficiency at low MOI.

## In vivo transduction and biodistribution of mutant AAV9

To assess in vivo transduction, BALB/c mice were intravenously injected with WT or mutant AAV9 vectors carrying the *NanoLuc* transgene. Vehicle contained PBS with no CpG adjuvant to resemble the AAV vector treatment given to patients in clinical settings. Luciferase expression levels in R312H-injected mice were comparable to WT AAV9, whereas R312Q-injected mice exhibited significantly lower expression after 30 days (Fig. 5C, D). These in vivo results are consistent with in vitro transduction assays, confirming that R312Q reduces transduction efficiency. Despite reduced transgene expression, all

**Table 1 | In silico analysis of mutated AAV9**

| Mutation | AA sequence | Mutation type | Percentile |
|---|---|---|---|
| Wild type | NNWGFRPKRLNFKLF | | 1.2 |
| F308D | NNWG**D**RPKRLNFKLF | Replacement | 29 |
| F308G | NNWG**G**RPKRLNFKLF | Replacement | 26 |
| F308N | NNWG**N**RPKRLNFKLF | Replacement | 23 |
| F308P | NNWG**P**RPKRLNFKLF | Replacement | 17 |
| F308E | NNWG**E**RPKRLNFKLF | Replacement | 16 |
| F308S | NNWG**S**RPKRLNFKLF | Replacement | 16 |
| F308Q | NNWG**Q**RPKRLNFKLF | Replacement | 14 |
| F308T | NNWG**T**RPKRLNFKLF | Replacement | 9.9 |
| R309D | NNWGF**D**PKRLNFKLF | Replacement | 23 |
| R309G | NNWGF**G**PKRLNFKLF | Replacement | 22 |
| R309N | NNWGF**N**PKRLNFKLF | Replacement | 14 |
| R309P | NNWGF**P**PKRLNFKLF | Replacement | 14 |
| R309S | NNWGF**S**PKRLNFKLF | Replacement | 11 |
| R309T | NNWGF**T**PKRLNFKLF | Replacement | 8.9 |
| R312D | NNWGFRPK**D**LNFKLF | Replacement | 18 |
| R312N | NNWGFRPK**N**LNFKLF | Replacement | 15 |
| R312G | NNWGFRPK**G**LNFKLF | Replacement | 11 |
| R312H | NNWGFRPK**H**LNFKLF | Replacement | 11 |
| R312Q | NNWGFRPK**Q**LNFKLF | Replacement | 9.6 |
| R312E | NNWGFRPK**E**LNFKLF | Replacement | 7.7 |
| F308^R309insD | NNWGF**D**RPKRLNFKLF | Insertion | 14 |
| F308^R309insG | NNWGF**G**RPKRLNFKLF | Insertion | 14 |
| F308^R309insN | NNWGF**N**RPKRLNFKLF | Insertion | 9 |
| R309^P310insG | NNWGFR**G**PKRLNFKLF | Insertion | 9.9 |
| F308del | NNWG_RPKRLNFKLF | Deletion | 19 |
| R309del | NNWGF_PKRLNFKLF | Deletion | 12 |

EMMP predictions were generated using MHC presentability data from the NetMHCII2.3 prediction method, which was the most up-to-date version available at the time of analysis.

groups maintained stable expression over 30 days (Fig. 5D), which correlates to 2.5 years in humans[16].

The biodistribution of mutated vectors in the mice was also analyzed by imaging each organ after harvesting them on day 30. AAV9 showed strong expression in the liver, heart, muscle, and thymus, which is consistent with previously published data[12,17]. Importantly, the R312H vector had a similar biodistribution to that of the WT control (Fig. 5E, F), indicating that this mutation does not alter tropism in mice. However, the biodistribution of R312Q was difficult to interpret due to the lower overall transgene expression that was more pronounced in the muscle and heart (Fig. 5E, F).

### Impact of mutations on T-cell activation and cytokine production

To determine whether these mutations effectively eliminated the immunogenic epitope, BALB/c mice were immunized with an equal dose of WT or mutant AAV9 in the presence of CpG adjuvant. Splenocytes were then restimulated with the corresponding peptides, and T-cell responses were assessed via ELISpot and flow cytometry. ELISpot analysis revealed that splenocytes from R312Q-immunized mice failed to produce IFN-γ and IL-2 upon stimulation with the mutated peptide (R312Q 102–103), whereas WT-immunized mice showed strong cytokine responses (Fig. 6A, B). Flow cytometry further confirmed that peptides 102–103 specifically activate CD4+ T cells, but not CD8+ T cells, to produce IFN-γ (Fig. 6C, D). Notably, the R312Q mutation abrogated this response, as IFN-γ production was absent in splenocytes stimulated with the mutant peptide. CD69 expression and TNF-α

production further validated the loss of CD4+ T-cell activation by the R312Q peptide (Fig. 6E, F). In contrast, the R312H mutation still induced CD4+ T-cell activation, albeit without significant cytokine production. These findings suggest that R312Q is more effective at eliminating the CD4+ T-cell epitope within peptides 102–103 compared to R312H, demonstrating the trade-off between transduction efficiency and immunogenicity.

### Reduced anti-AAV9 antibody response in mutant AAV9-immunized mice

To investigate whether T-cell deimmunization also reduced humoral immunity, serum samples from immunized mice were collected on days 5, 16, and 29 post AAV injection for anti-AAV9 antibody quantification via ELISA. All groups developed detectable anti-AAV9 antibodies by day 5, with titers increasing significantly by day 16 and remaining stable through day 29. Importantly, mice immunized with the mutant AAV9 variants exhibited significantly lower antibody titers compared to WT AAV9 (Fig. 7A, B), indicating that mutations within peptides 102–103 reduce the generation of anti-AAV antibodies.

To further evaluate the functional significance of reduced binding antibodies, we assessed the neutralizing capacity of sera from immunized mice. Neutralizing antibody analysis revealed that both mutant vectors, particularly R312H, showed significantly reduced neutralizing activity compared to wild-type AAV9 (Fig. 7C, D), indicating that epitope modification can reduce functionally relevant immune responses that could impact vector re-administration.

### Discussion

Our study demonstrates that targeted modification of CD4+ T-cell epitopes in the AAV9 capsid can effectively reduce immune responses while maintaining key aspects of vector function. By identifying immunodominant T-cell epitopes in BALB/c mice and engineering site-specific mutations, we showed that the R312Q mutation significantly diminishes CD4+ T-cell activation and cytokine production, which results in reduced anti-AAV9 antibody formation. These findings offer proof of concept for the notion that rational disruption of class II MHC-restricted epitopes is a feasible strategy to mitigate adaptive immune responses to AAV vectors. Our studies also provide a bioinformatic tool provided to the public that can effectively assist with mutation selection and vector design.

AAV-mediated gene therapy has been limited by host immune responses, particularly T-cell activation against capsid-derived peptides presented by MHC class molecules[18,19]. Our previous studies have identified immunogenic regions in AAV capsids using human donor samples, highlighting the potential for rational epitope modification to improve therapeutic outcomes[12]. Our results align with these findings, demonstrating that disrupting a single immunodominant epitope can significantly alter immune recognition. Interestingly, the epitope we identified in mice is proximal to, but distinct from, an epitope previously reported in human samples. This highlights the importance of considering species-specific differences when developing strategies to reduce vector immunogenicity. Of note, proximity or commonality between human and murine MHC-II-presented epitopes does not occur very frequently but is not rare. Previous work by us and others found common or proximal immunodominant epitopes between humans and BALB/c mice in bacterial toxins, human cytokine or human receptors[20–22].

The epitope in amino acids 304–321 can be found in all three viral proteins VP1, VP2, and VP3 in the non-variant regions[13]. While the murine T cell epitopes in AAV9 have not been described yet, previous work focusing on AAV2 identified a murine CD8 epitope in the BALB/c strain in amino acids 306–320 of AAV2 (FRPKRLNFKLFNIQV), which corresponds to amino acids 308–322 in AAV9[23]. AAV2 and AAV9 have 100% homology of the amino acids in this amino acid sequence. The high proximity of the T cell epitope in AAV2 and AAV9 further confirms

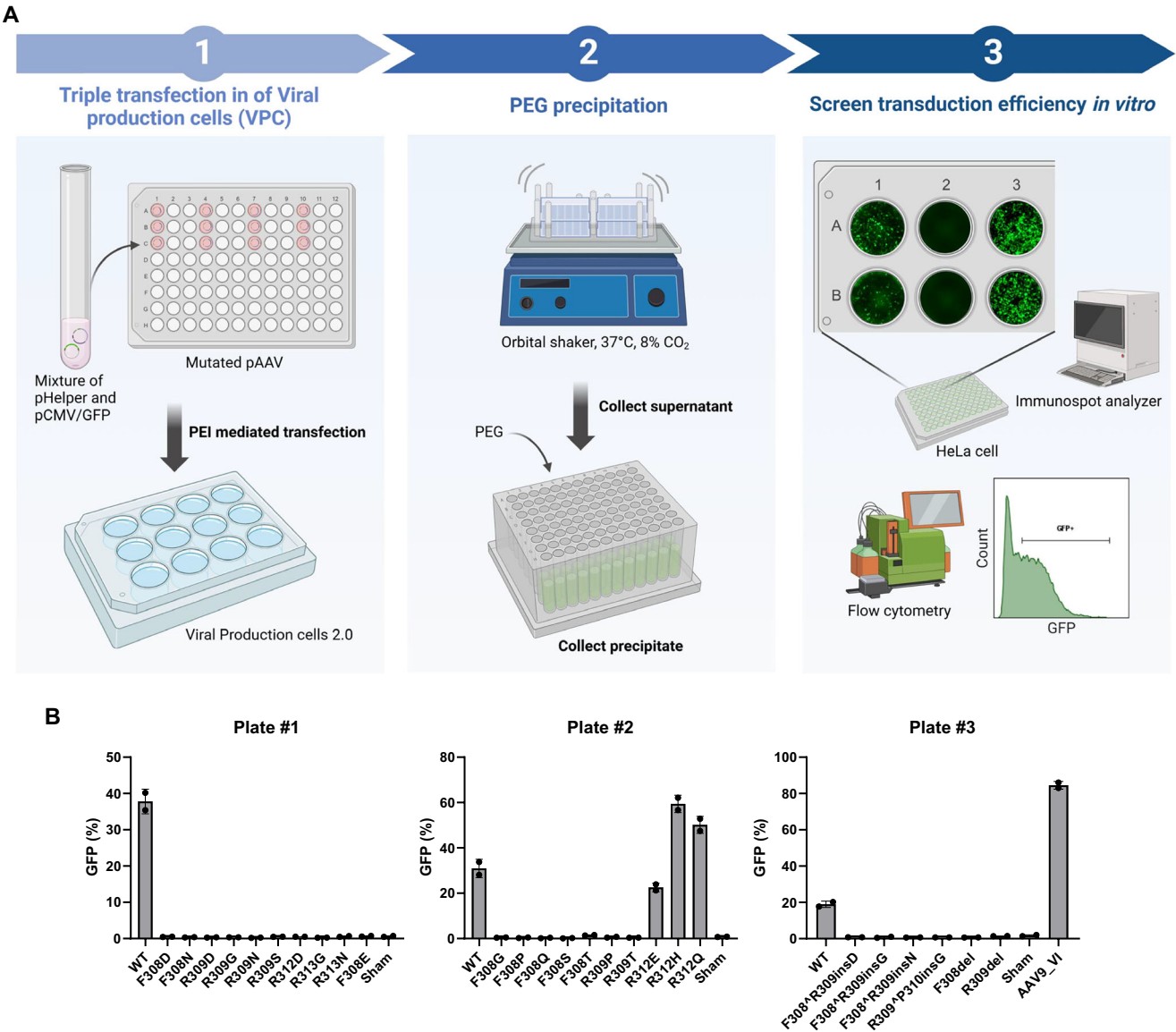

**Fig. 4 | Selection of mutated AAV candidates that maintain transduction efficiency using a semi-high-throughput screening method. A** Overview of engineered AAV production with high-throughput screening. Viral production cells (VPCs) were transfected using PEI with three plasmids (*pHelper, pCMV/GFP*, and mutated *pAAV*) and incubated for 3 days at 37 °C with 8% $CO_2$ and 110 rpm shaking. The supernatant was harvested, and rAAV vectors were precipitated by adding ¼ volume of 40% PEG 8000. After overnight incubation, the rAAV particles were collected by centrifugation, resuspended in the final buffer, and directly applied to HeLa cells. Following a 3-day incubation, GFP-expressing HeLa cells were analyzed by ELISpot reader and flow cytometry. Created in BioRender. Bing, S. (2026) https://BioRender.com/c76n313. **B** The percentage of GFP-expressing cells among live cells was measured by flow cytometry. Each sample was prepared in duplicate, with one replicate in each of two separate plates. Values are presented as mean ± SD. Source data are provided as a Source data file.

the significance of this epitope. Interestingly, a similar investigation of T cell epitopes in AAV8 only identified a single epitope, which did not overlap with the epitope in 304–321 despite the equally high sequence homology[23].

For modifying immunogenic regions, we developed the EMMP pipeline that predicts the binding affinity of all possible single-residue substitutions within a given epitope on MHC. EMMP systematically evaluates binding to multiple MHC alleles, identifies candidate mutations that reduce MHC presentability, and visualizes results to support interpretation. By integrating these predictions with experimental screening, we successfully identified R312 as a critical anchor residue for MHC class II interaction and demonstrated that mutations at this site could reduce immune response.

The EMMP pipeline offers a significant advancement in the systematic application of MHC-binding prediction methods by automating comprehensive mutation analysis. While databases like IEDB support peptide-MHC binding predictions[24], prediction algorithms such as NetMHC require manual evaluation of individual mutations. EMMP overcomes these limitations by automating the application of prediction of replacements, insertions, and deletions, calculating median and minimum MHC presentability scores across multiple alleles. This capability is especially useful when modifying promiscuous epitopes, where a mutation may reduce binding to one allele but increase it for another. This automation represents a substantial methodological advancement that reduces both time investment and potential errors in the rational design of biological products.

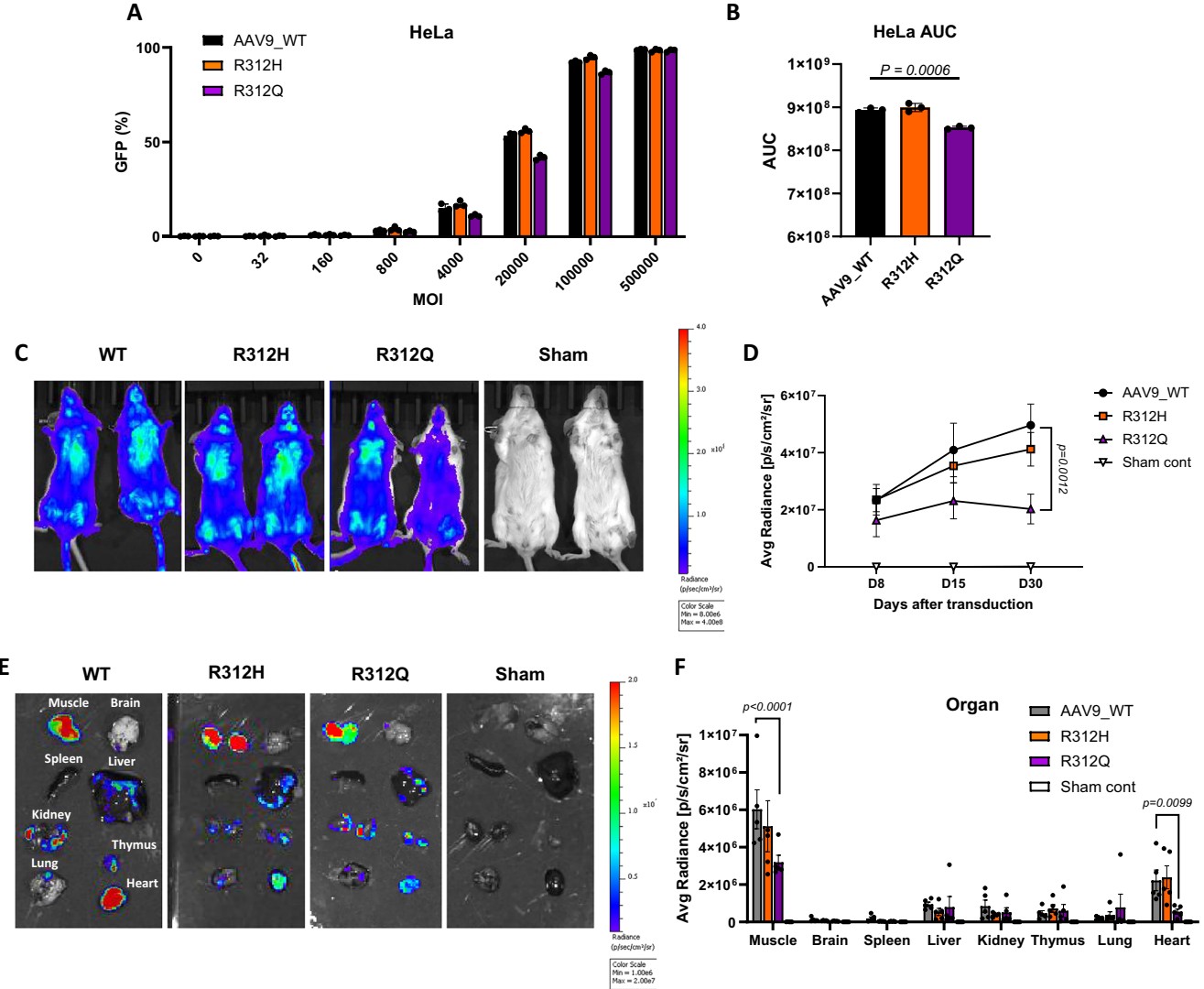

**Fig. 5 | Transduction efficiency and biodistribution of mutated AAV vectors.**
**A**, **B** HeLa cells were transduced with the AAV9, R312H, and R312Q vectors at an indicated MOI. The percentage of GFP-positive cells was determined by flow cytometry. **B** Four-parameter curve fit was calculated for each vector, and the area under the curve (AUC) was calculated. Data are shown as a representative result from three independent experiments with similar results. For each experiment, $n = 3$ independent biological replicates were used. Each bar shows the mean ± SD. *P* values were determined by one-way ANOVA with Tukey's multiple comparisons test. **C**–**F** Female BALB/c mice (10 weeks, $n = 5$/group) were injected with 1E11 vg of AAV9, mutated AAV9, or sham control intravenously. **C** Representative image of NanoLuc expression in the mice 8 days after vector administration.

**D** Quantification of NanoLuc signal in vector-injected mice on days 8, 15, and 30 days after vector administration. Average radiance was measured at 6 ± 1 min after substrate injection. Symbols represent AAV9_WT (black circles), R312H (orange squares), R312Q (purple triangles), and Sham control (open inverted triangles). Each bar shows the mean ± SEM. Representative images (**E**) and quantification (**F**) of luciferase signal in various organs of vector-injected mice on day 30. Bar colors represent AAV9_WT (black), R312H (orange), R312Q (purple), and Sham control (white). Each bar shows the mean ± SEM. *P* values were determined by two-way ANOVA with Dunnett's multiple comparisons test. Source data are provided as a Source data file. MOI multiplicity of infection, AUC area under the curve, Sham sham control.

Our application of EMMP to the AAV9 capsid epitope revealed that positions 308 (F), 309 (R), and 312 (R) are key contributors to MHC binding, consistent with the anchor residue model in T-cell immunology. Functional screening of EMMP-predicted variants confirmed that R312Q is indeed more effective at reducing immune activation. The differential effects observed across substitutions at this position, such as the reduced efficacy of R312E, suggest that side chain charge and structure play critical roles not only in presentability but also in capsid integrity and function. The wild-type arginine may participate in electrostatic interactions essential for capsid assembly or receptor binding, and these interactions may be better preserved by uncharged polar residues like histidine or glutamine than by negatively charged glutamate.

Although the R312Q mutation showed slightly reduced transduction efficiency in vitro and in vivo compared to WT AAV9, the significant reduction in immune responses could outweigh this minor decrease in gene transfer. The trade-off between transduction efficiency and immunogenicity is a crucial consideration in vector design. The reduced anti-AAV antibody production in mice immunized with mutated vectors is particularly promising for clinical applications. Lower antibody titers could potentially allow for more effective initial treatment and may facilitate vector re-administration, a major challenge in AAV-mediated gene therapy.

Despite these promising results, several limitations should be acknowledged. First, while EMMP relies on algorithms with strong predictive power, all computational models have inherent

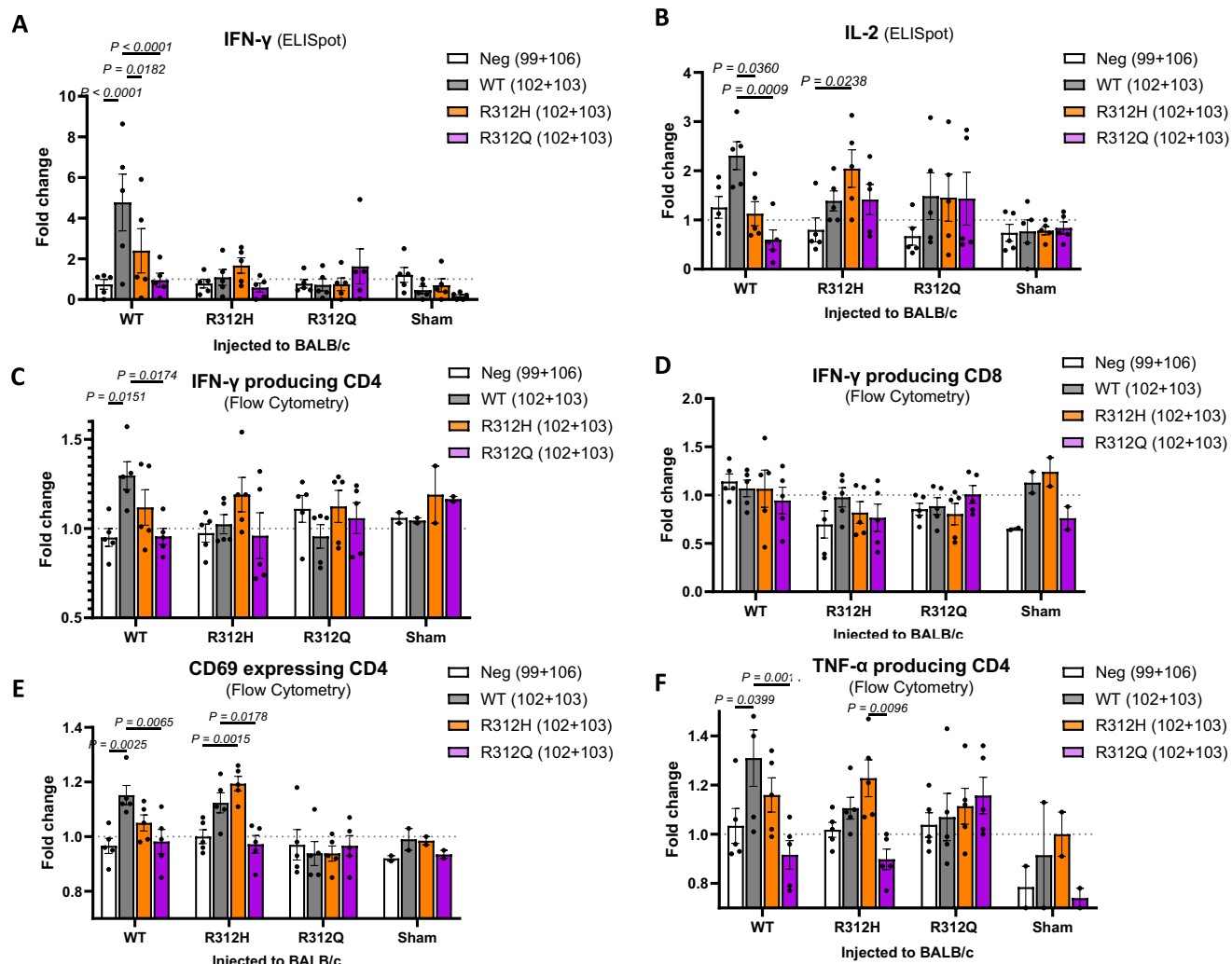

**Fig. 6 | The R312Q AAV9 mutant does not activate the CD4 T cells to produce IFN-γ and IL-2.** Female BALB/c mice (8 weeks, *n* = 5/group) were injected with 1E11 vg of AAV9, mutated AAV9, or sham control (indicated in *x*-axis) along with 5 μM of CpG intraperitoneally. On day 9, mice were euthanized, and splenocytes were isolated. Cells were restimulated with the indicated peptides, and the IFN-γ (**A**) and IL-2 (**B**) producing cells were measured by ELISpot assays. IFN-γ-producing cells were measured within the CD4 (**C**) or CD8 (**D**) T gated cells using intracellular flow cytometry staining. CD69+ cells (**E**) and TNF-α (**F**) producing cells within the CD4 T gate were measured. Fold changes were calculated by comparing peptide-stimulated responses to medium-only control (no peptide) for each treatment group. The *X*-axis indicates the type of AAV vector injected into mice. Color bars indicate the peptides used for restimulation: irrelevant peptide (neg (99 + 106)), wild-type peptide sequence (WT (102 + 103)), or mutant peptide sequences (R312H (102 + 103), R312Q (102 + 103)). Values are presented as mean ± SEM. Significance was calculated using a one-way ANOVA with Dunnett's multiple comparisons test. Source data are provided as a Source data file. Neg negative control, WT wild type.

limitations[25,26]. An important limitation of our computational pipeline is the dependence on the choice of MHC-binding prediction algorithm. Our study utilized NetMHCII2.3, which was the most current and IEDB-recommended method at the time of our experimental work (2021–2022). However, comparison with the more recently recommended NetMHCIIPan4.0 EL method reveals substantially different predictions for our selected mutations. While NetMHCII2.3 predicted R312H and R312Q as having reduced MHC presentability, NetMHCII-Pan4.0 EL suggests these mutations would retain strong MHC presentability. This finding reinforces our core argument: the primary value of the EMMP framework is not in the absolute accuracy of any single predictive model, but in providing a systematic, automated pipeline for candidate generation. This divergence underscores the critical importance of experimental validation to bridge the gap between computational prediction and actual biological reality. Future applications of the EMMP pipeline should incorporate multiple prediction methods and prioritize empirical validation to ensure robust candidate selection across different algorithmic approaches for MHC

presentability prediction. Prediction accuracy may also decrease for novel peptide types or less-studied MHC alleles due to gaps in training data[25]. Importantly, EMMP and the algorithms it utilizes are designed to predict peptide-MHC binding rather than actual immunogenicity. Strong predicted binding is necessary for T-cell recognition but not sufficient to guarantee an immune response[27]. Actual immunogenicity depends on multiple additional factors, including antigen processing and presentation efficiency, T-cell receptor repertoire diversity, central and peripheral tolerance mechanisms, peptide abundance, co-stimulatory signals, and the broader immunological context[26,28]. Consequently, peptides with moderate predicted binding may still elicit immune responses, while others with strong predicted binding may not[29]. Training data biases can further skew predictions if some MHC alleles are underrepresented[30], limiting the reliability of predictions across diverse genetic backgrounds[31]. This distinction highlights that EMMP serves as a screening tool to prioritize candidate mutations but cannot definitively predict immunogenicity reduction, underscoring the necessity of experimental validation. Additionally, our semi-high-

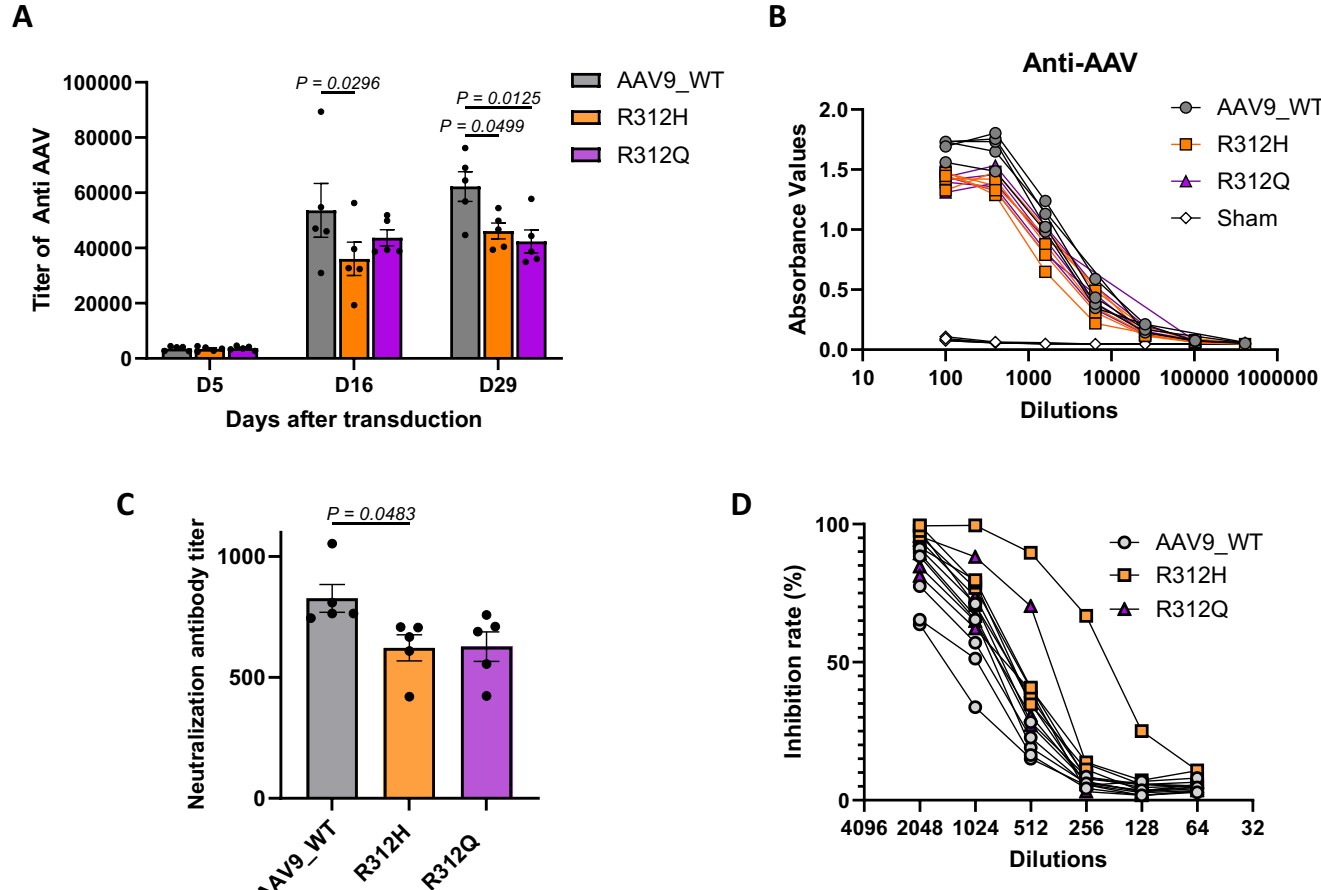

**Fig. 7 | AAV9 mutants reduce anti-AAV9 antibody responses.** BALB/c mice (10 weeks, *n* = 5/group) were injected with 1E11 vg of AAV9, mutated AAV9, or sham control intravenously. **A** On days 5, 16, and 29, serum was collected for anti-AAV titer. Anti-AAV was not detected in all mice before the AAV injection. **B** Anti-AAV in individual mice at various dilutions on day 16. **C, D** Neutralizing activity of serum samples against AAV9 or its mutants. Neutralization titers were determined based on GFP transgene expression in HeLa cells 3 days after transduction. **D** Inhibition of GFP transduction efficiency by sera from mice. Bar colors represent AAV9_WT (gray), R312H (orange), and R312Q (purple). Individual lines and symbols represent AAV9_WT (gray circles), R312H (orange squares), R312Q (purple triangles), and Sham control (open diamonds). Each bar shows the mean ± SEM. *P* values were determined by one-way ANOVA with Tukey's multiple comparisons test. Source data are provided as a Source data file. Sham sham control.

throughput screening provided relative transduction efficiencies without precise MOI determination, which will require more detailed characterization.

Although the specific epitope substitutions evaluated here may not directly translate to clinical settings, this study provides a mechanistic demonstration and methodological foundation for rational immunogenicity reduction. Additionally, while we focused on a specific murine epitope, human MHC alleles exhibit substantial polymorphism, potentially necessitating more optimization steps for clinical applications. While we acknowledge the challenges of translating findings from a single mouse epitope to the complex multi-epitope landscape of human immune responses, our EMMP approach is designed to address HLA diversity systematically. By evaluating mutations across 27 representative HLA alleles that provide maximal population coverage[32], EMMP can identify substitutions that consistently reduce MHC presentability across diverse genetic backgrounds. The broader applicability of EMMP is demonstrated by its successful application to adenoviral hexon epitopes, where computationally predicted mutations corresponded to empirically validated modifications that have been patented for clinical use (Supplementary Fig. 4). This suggests that EMMP can serve as a general framework for rational deimmunization across multiple viral vector platforms.

In conclusion, EMMP offers a practical and efficient method for identifying mutations that reduce MHC presentability. By applying this strategy to the AAV9 capsid, we demonstrated that a single-point mutation can meaningfully reduce immune responses in vivo. Future studies should focus on combining epitope modification with other strategies, such as capsid engineering for enhanced tissue specificity or reduced liver tropism (for indications targeting organs other than the liver), to further improve the safety and efficacy of AAV-mediated gene therapy. By strategically modifying T-cell epitopes, we can potentially enhance the efficacy and broaden the applicability of AAV-mediated gene therapy, bringing us closer to realizing its full potential in treating a wide range of genetic disorders.

## Methods
### Ethical statement
The research in this paper complies with all ethical regulations set by the FDA White Oak Consolidated Animal Care and Use Committee. Animals were maintained and treated in strict accordance with FDA institutional guidelines and the standards contained in the Guide for the Care and Use of Laboratory Animals (The Guide, 8th Edition) from the National Institutes of Health. Animal protocols were approved by the FDA White Oak Consolidated Animal Care and Use Committee (protocols 2020−18). Animal facilities were accredited by the

Association for Assessment and Accreditation of Laboratory Animal Care International. For vector production, Viral Production Cells 2.0 (A49784, Thermo Fisher Scientific) were utilized. For subsequent in vitro transduction assays, HeLa (CCL-2, ATCC) and HEK293T (CRL-3216, ATCC) cell lines were employed. All cell lines were obtained directly from the respective vendors and were authenticated by the manufacturers prior to purchase. As these cells were used within a minimal number of passages upon receipt, no further authentication was performed in our laboratory.

### Peptide synthesis and recombinant vector production

A total of 242 15-mer peptides, overlapping by 12 amino acids, were synthesized by GenScript Biotech (Piscataway, NJ, USA). These peptides encompassed the entire sequence of *VP1* from the AAV9 capsid. The peptides were confirmed to have a purity of >95% through high-performance liquid chromatography. Lyophilized peptides were reconstituted in dimethyl sulfoxide at a concentration of 50 mg/mL and subsequently diluted in RPMI media. For the initial screening, the peptides were arranged into a matrix of 15 vertical pools and 16 horizontal pools, with each pool containing 15 or 16 peptides (Fig. 1A).

AAV production was carried out using the triple transfection method as previously described[33], with minor modifications. Briefly, Viral Production Cells 2.0 (A49784, Thermo Fisher Scientific) were transfected and harvested 72 h post-transfection. For transfection, the plasmid, *pAAV2/9n* (112865, Addgene, Watertown, MA), *pHelper* (340202, Cell Biolabs, San Diego, CA), *pscAAV-GFP* (AAV-410, Cell Biolabs), and *pAAV-NanoLuc-HT* (Promega, Madison, WI) were used. Following three freeze-thaw cycles, sonication, and benzonase treatment, AAVs were purified through two successive rounds of ultracentrifugation using iodixanol gradients. For the sham control, all procedures were performed identically except for the omission of the *Rep/Cap* plasmid. Capsid proteins in the vectors were quantified using AAV9 ELISA kits (Progen, Wayne, PA, USA). Vector genome copy titers were determined by quantitative PCR (qPCR) targeting the inverted terminal repeat (ITR) region[34]. The specific sequences for the ITR primers were as follows: forward, 5′-GGAACCCCTAGTGATGGAGTT-3′; reverse, 5′-CGGCCTCAGTGAGCGA-3′. Briefly, AAV9 samples were treated with DNase I to remove residual plasmid DNA, followed by serial dilution. qPCR was performed using SsoAdvanced Universal SYBR Green Supermix (Bio-Rad, Cat# 1725272, Hercules, CA, USA) on a Bio-Rad iCycler iQ Multicolor Real-Time PCR Detection System. qPCR was performed with an initial denaturation at 98 °C for 30 s, followed by 39 cycles of 98 °C for 30 s and 58 °C for 30 s. Specificity was verified via melt curve analysis (65–95 °C), and titers were determined using a standard curve.

### Analytical assessment of full, partial, and empty aav particles through charge detection mass spectrometry (CDMS)

CDMS analysis of the engineered vectors (AAV9_WT, R312H, and R312Q) was performed by taking 25 µL of each sample and directly performing buffer exchange into 200 mM aqueous ammonium acetate (Sigma-Aldrich, A2706) with 0.01% pluronic F-68 (Thermo Fisher Scientific, 24040032) using Micro Bio-Spin P-6 gel columns (Bio-Rad, 7326221). Mass analysis was performed using a prototype Charge Detection Mass Spectrometer (Waters Corporation) with an electrostatic linear ion trap that is based on the system built by Megadalton Solutions, which has been described previously[35,36]. A prototype nanoelectrospray ionization (nESI) source (Waters Corporation) using 5 µm ID glass emitters (World Precision Instruments, TIP5TW1) generated positive ions with voltages between 1.9 and 2.1 kV. The spectra were collected until approximately 3000 ions were captured within the mass range of 3–5 MDa, with a total acquisition time of approximately 30 min. Three ($n = 3$) technical replicates were acquired. Signal processing and data visualization were performed using prototype software developed in-house. The mass range of the CDMS system was

calibrated using L-glutamate dehydrogenase (GDH) from bovine liver (Sigma-Aldrich, G7882). To generate this calibration, the $m/z$ peaks in the GDH spectra were plotted against the theoretical $m/z$ values of GDH to give a correction factor that was used to accurately determine the mass and charge from the CDMS system.

### Mice and immunization protocols

Six-to-eight-week-old female BALB/c mice were purchased from The Jackson Laboratory (Bar Harbor, ME). Mice were bred and housed in the FDA AAALAC-accredited, pathogen-free animal facility. Mice were housed in standard cages with 1 breeding pair or up to 5 single sex mice per cage and a 12/12 light/dark schedule and fed on commercial 5P76 Prolab Isopro RMH 3000 diet. Animal studies were reported according to ARRIVE guidelines, and no animals were excluded from the study. For all experiments, female BALB/c mice were used to ensure experimental consistency and to minimize physiological variability. They exhibit lower levels of aggression compared to males, which facilitated stable co-housing of multiple experimental groups and reduced stress-induced variables. For the identification of epitopes, mice ($n = 18$) were immunized with 1E11 vg of AAV9 and 5 µM of CpG intraperitoneally. Nine to ten days after the immunization, the mice were sacrificed, and their spleens were removed. Splenocytes were isolated by standard methods, and single-cell suspensions, depleted of red blood cells, were prepared from freshly isolated splenocytes in culture medium (RPMI 1640 medium supplemented with 10% v/v fetal bovine serum, 100 U/mL penicillin/streptomycin (Thermo Fisher Scientific), 2 mM L glutamax (Thermo Fisher Scientific), 50 µM 2-mercaptoethanol (Thermo Fisher Scientific) and 10 mM HEPES (Thermo Fisher Scientific)). Each mouse was processed independently, and ELISpot assays were performed in triplicate for each individual mouse. Data from all 18 mice were then pooled for statistical analysis. Cells were screened for reactivity against AAV-specific peptide pools or individual peptides by ELISpot assay or flow cytometry.

### ELISpot assay

The secretion of IL-2 and IFN-γ was evaluated using an ELISpot assay, performed according to the manufacturer's instructions (Mabtech, Cincinnati, OH, USA). Splenocytes were plated at a density of 300,000 cells per well and stimulated with either peptide pools or individual peptides (10 µg/mL) in plates pre-coated with anti-mouse IL-2 or IFN-γ antibodies. Negative controls were treated with medium alone, while positive controls were stimulated with concanavalin A (ConA, MilliporeSigma). After 24 h of incubation, spots were developed using biotin-conjugated anti-IL-2 or anti-IFN-γ antibodies (Mabtech, Nacka Strand, Sweden), streptavidin-alkaline phosphatase (Mabtech), and the BCIP/NBT substrate (KPL, Thermo Fisher Scientific). Spot-forming cells were enumerated using ImmunoSpot 7.0 software (Cellular Technology Limited, Cleveland, OH, USA). Responses were considered positive if their magnitude exceeded 2-fold (for IFN-γ) or 3-fold (for IL-2) compared to the background level of negative controls.

For experiments involving the isolation of CD4 or CD8 T cells, the respective subsets were purified by positive selection using magnetic beads and LS columns (Miltenyi Biotec, Bergisch Gladbach, Germany) in accordance with the manufacturer's protocol.

### Flow cytometry

Splenocytes were restimulated with peptides (10 µg/mL) for 24 h at 37 °C. For negative controls in T-cell assays, peptides 99 and 106 were selected from the 242-peptide library based on their low predicted MHC class II binding affinity and proximity to the immunogenic region (peptides 102 + 103). These peptides serve as irrelevant controls to account for non-specific T-cell activation in the same sequence region. To block cytokine secretion in cell cultures, GolgiPlug/GolgiStop (BD Biosciences, San Jose, CA, USA) was added 4 h prior to cell harvesting and staining. Cells were stained for surface markers CD3 (1:1000, clone

500A2, BioLegend, cat. 560771, RRID: AB_2564590), CD4 (1:1000, clone RM4-5, BioLegend, cat. 100528, RRID: AB_312729), CD8 (1:1000, clone 53-6.7, BioLegend, cat. 100706, RRID: AB_312745), and CD69 (1:1000, clone H1.2F3, BioLegend, cat. 104512, RRID: AB_493564). After washing, cells were fixed and permeabilized using Cytofix/Cytoperm solution (BD Biosciences) and subsequently stained for intracellular cytokines IFN-γ (1:500, clone XMG1.2, BioLegend, Cat. 505808, RRID: AB_315402) and TNF-α (1:500, clone MP6-XT22, BioLegend, Cat. 506328, RRID: AB_2562902). Data acquisition was performed on a Cytek Aurora (Cytek Biosciences, Fremont, CA, USA), and analysis was conducted using FlowJo software (version 10.8, Treestar, Ashland, OR, USA).

## In silico MHC-II binding prediction

Peptide binding affinities for MHC class II were predicted using the binding prediction tool available at the IEDB[19]. Binding predictions were performed for each 15-mer peptide against BALB/c alleles (H2-IAd and H2-IEd). The percentile rank was used to standardize comparisons across different prediction models, with lower percentile values indicating higher binding affinities.

## EMMP pipeline for the Replacement Algorithm

This method computes the binding affinity of peptides with amino acid substitutions, where each residue in the original peptide sequence is systematically replaced with all 20 possible wild-type amino acids. The algorithm iterates over each position in the peptide, generating modified sequences by substituting the native amino acid with each of the 20 alternatives. Then, the EMMP utilizes an MHC prediction tool is then employed to calculate the promiscuous MHC presentability for each modified peptide, using the following representation: Binding Affinity (P) = F (P, Alleles), where F denotes the MHC prediction function, P represents the modified peptide sequence, and Alleles corresponds to the list of selected alleles. As detailed in Algorithm 1, this process involves generating a Fasta file for each modified peptide, which is then used as input for the IEDB MHC prediction tool. The predicted binding affinity analysis is subsequently retrieved and stored in a data frame, facilitating further analysis in downstream stages. By exhaustively exploring all possible amino acid substitutions at each possible position, this pipeline provides a comprehensive assessment of the impact of residue replacements on peptide binding affinity.

**Algorithm 1**. Replacement Algorithm
1: function CalculateReplacementBindingAffinity(P, AAs, Alleles)
2: $n = length(P)$
3: for $i = 0$ to $n - 1$ do ▷ iterate over each position in the sequence
4: for $\alpha \in AAs$ do ▷ iterate over each amino acid
5: $P' = (p_1, p_2, ..., p_i\text{ -1}, \alpha, p_i + 1, ..., p_n)$
6: Add modified peptide sequence P' to Fasta file
7: end for
8: $BindingAffinity(P') = F (P', Alleles)$
9: Store the predicted binding affinity
10: end for
11: end function

## EMMP pipeline for the Insertion Algorithm

The insertion method computes the binding affinity predictions of peptides with insertions by iterating over each position in the given peptide sequence. At each position, an amino acid is inserted between the current and the next amino acids, generating a new peptide variant. As outlined in Algorithm 2, this process involves inserting a novel amino acid at each position, resulting in a Fasta file that includes the modified peptide sequences. The algorithm then utilizes the IEDB MHC prediction tool to retrieve the predicted binding affinity for each variant, storing the results in a data frame for further analysis.

**Algorithm 2**. Insertion Algorithm
1: function CalculateInsertionBindingAffinity(P, AAs, Alleles)
2: $n = length(P)$
3: for $i = 0$ to $n - 2$ do ▷ iterate over each position in the sequence
4: for each possible insertion point do ▷ iterate over each amino acid
5: $\alpha = choose\ Amino\ Acid\ from\ AAs$
6: $P' = (p_1, ..., p_i\text{ -1}, \alpha, p_i + 1, ..., p_n)$
7: Add modified peptide sequence P' to Fasta file
8: end for
9: $BindingAffinity(P') = F (P', Alleles)$
10: Store the predicted binding affinity
11: end for
12: end function

## EMMP pipeline for the Deletion Algorithm

The deletion algorithm (Algorithm 3) systematically iterates over each position in the peptide sequence, generating a modified peptide sequence P' by deleting the amino acid at each respective position. The resulting modified sequences are then appended to a Fasta file. Then, the algorithm calls the MHC prediction pipeline to compute the binding affinity of these modified sequences for all selected alleles, storing the calculated affinities in a data structure for further analysis and evaluation.

**Algorithm 3**. Deletion Algorithm
1: function CalculateDeletionBindingAffinity(P, Alleles)
2: $n = length(P)$
3: for $i = 0$ to $n - 1$ do ▷ iterate over each position in the sequence
4: $P' = (p_1, ..., p_i\text{ -1}, \alpha, p_i + 1, ..., p_n)$ ▷ delete $p_i$ position
5: Add modified peptide sequence P' to Fasta file
6: end for
7: $BindingAffinity(P') = F (P', Alleles)$
8: Store the predicted binding affinity
9: end function

## Semi-high-throughput AAV production and purification in a 96-well plate system

Plasmids of each mutated AAV were distributed into a 96-well plate, leaving two columns empty between different variants. A mixture containing the helper plasmid and *GFP* plasmid was added using a multi-pipette. Subsequently, polyethyleneimine (PEI, Polysciences, Warrington, PA, USA) at a concentration of 1 mg/mL was added at a 2:1 PEI-to-DNA ratio, followed by gentle pipetting and incubation for 15 min. The plasmid-PEI mixture was then transferred into a 12-well plate containing 1.5 mL of viral production media (Thermo Fisher Scientific, Waltham, MA, USA) supplemented with glutamate and Viral Production Cells 2.0 (Thermo Fisher Scientific) at a density of $3 \times 10^6$ cells/mL. The plate was covered with breathable AeraSeal (A9224, MilliporeSigma) and incubated at 37 °C with 8% $CO_2$ while shaking at 110 rpm for 3 days. After incubation, the culture supernatant was collected by centrifugation at $1000 \times g$ for 10 min and transferred into a deep 96-well plate using an adjustable tip spacing pipette. PEG/NaCl solution (40% PEG 8000 + 2.5 M NaCl) was added at a ratio of ¼ of the total volume using a multi-pipette, followed by shaking at 4 °C overnight. The following day, the plate was centrifuged at $2500 \times g$ for 1 h, and the supernatant was discarded. The remaining rAAV particles were resuspended in 100 µL of final buffer (0.001% Pluronic F-68, 10 mM $MgCl_2$, 35 mM NaCl, and 5% glycerol in PBS).

## Evaluation of rAAV transduction efficiency in HeLa cells

HeLa cells ($2 \times 10^4$, CCL-2, ATCC, Manassas, VA, USA) were seeded in 50 µL of DMEM (Thermo Fisher Scientific) supplemented with 2% FBS (Hyclone, Logan, UT, USA) and incubated for 2 h. Subsequently, rAAV particles obtained from the previous step were added at 50 µL per well

to transduce the HeLa cells. The following day, 100 μL of DMEM supplemented with 10% FBS and 1% penicillin/streptomycin (Thermo Fisher Scientific) was added, and the cells were further incubated for an additional 2 days. GFP-expressing cells were imaged using ImmunoSpot 7.0 software (Cellular Technology Limited, Cleveland, OH, USA). After imaging, cells were detached from the plate using 0.05% trypsin/EDTA (Thermo Fisher Scientific) and transferred to a U-bottom 96-well plate. The cells were then washed with PBS, and the resulting cell pellet was resuspended in PBS + 0.5% BSA (MilliporeSigma, Burlington, MA, USA) buffer containing propidium iodide (PI, MilliporeSigma) at a 1:1000 dilution. Data acquisition was performed using a Cytek Aurora flow cytometer (Cytek Biosciences, Fremont, CA, USA), and data analysis was conducted using FlowJo software (Treestar, Ashland, OR, USA).

### DNA mutagenesis
Using a custom-designed computational pipeline to systematically identify and catalog potential mutations in the submitted sequences. This approach enabled us to comprehensively evaluate the effects of various mutational events, including amino acid substitutions, insertions, and deletions. From this analysis, we selected a total of 26 mutations for further investigation. Plasmid mutagenesis was carried out by Gene Universal by site-directed mutagenesis (Newark, DE, USA). The accuracy of the nucleotide changes was confirmed by DNA sequence analysis immediately after production and again following plasmid expansion.

### In vitro transduction assay
HeLa or HEK293T (CRL-3216, ATCC) cells were seeded in 96-well plates at a final density of $2 \times 10^4$ cells/well in DMEM (Thermo Fisher) supplemented with 10% fetal bovine serum (FBS, MilliporeSigma) and penicillin/streptomycin. After 24 h, the media were replaced with DMEM containing 2% FBS, and rAAV vectors were added at varying multiplicities of infection (MOIs). At 72 h post-transduction, GFP expression was analyzed using flow cytometry. For the detection of NanoLuc luciferase, furimazine (Promega) was added to the culture medium 24 h post-transduction, and luminescence was immediately measured using a luminometer.

### In vivo bioluminescence imaging of NanoLuc
NanoLuc-expressing rAAV vectors were diluted in 200 μL of PBS and administered intravenously via the tail vein at a dose of 1E11 vector genomes (vg) per mouse (5 mice per group, 6–8 weeks, female). On days 8, 15, and 30 post-injection, mice were injected with 0.44 μmol of fluorofurimazine (Promega), anesthetized with isoflurane, and imaged 6 ± 1 min later using an IVIS Spectrum imager (PerkinElmer, Waltham, MA, USA). On day 30, mouse organs were harvested, rinsed in PBS, and analyzed with the IVIS imager. Signal quantification in specific regions of interest was corrected for background by subtracting signals obtained from the corresponding organs of sham controls.

### Anti-AAV titer
Anti-AAV antibodies in mouse serum were quantified using a standard ELISA. Ninety-six-well ELISA plates (Thermo Fisher) were coated overnight with wild-type (WT) or mutated (R312H or R312Q) AAV9 in boric acid buffer at a final concentration of $5 \times 10^{10}$ viral particles (vp) per well. The plates were blocked with blocking buffer (PBS containing 3% bovine serum albumin) for 2 h at room temperature. Serum samples were incubated with the plates, followed by anti-mouse IgG antibodies conjugated to HRP (Jackson Laboratory, Bar Harbor, ME, USA, cat. 115-035-166, RRID:AB_2338511) for 1 h. Detection was performed using enzymatic development with 3,3′,5,5′-tetramethylbenzidine (TMB). Titer values for all ELISA assays represent the dilution factor at

which the ΔO.D signal is similar to the signal obtained by negative samples. Titer was interpolated using a 4-parameter curve fit in GraphPad Prism. The cut point that represents the ΔO.D of negative samples was calculated by the sum of 3 x the standard deviation and the average of pre-treatment sera ΔO.D values at the highest concentration.

For neutralizing antibody assessment, serum samples collected on day 16 were serially diluted and pre-incubated with either wild-type AAV9 or mutated AAV vectors (R312H or R312Q) for 1 h at room temperature. The AAV-serum mixtures were then applied to HeLa cells seeded in 96-well plates for transduction analysis. After 72 h, GFP expression was analyzed by flow cytometry to determine the neutralizing capacity of the sera. Neutralization titers were defined as the serum dilution that reduced transduction efficiency by 50% compared to control (no serum). The percentage inhibition of transduction was calculated as: [(control transduction − test transduction) / control transduction] × 100.

### Statistics and reproducibility
Statistical analyses were conducted using GraphPad Prism version 10.2 (GraphPad Software, San Diego, CA, USA). No statistical method was used to predetermine sample size, which was chosen based on prior experience and standard practice in the field to ensure adequate power for comparison. In Fig. 7A, B, data from certain wells were excluded from the final analysis because of technical issues during the plate washing process, where a clogged nozzle prevented proper washing. Aside from these specific instances, no other data were excluded from the analyses. The investigators were blinded to group allocation during data collection and outcome assessment to minimize bias. Samples were coded by a separate researcher so that the investigators performing the assays were unaware of the specific group identities until the final analysis was completed. All experiments were independently repeated as described in the figure legends, and key findings were validated across multiple biological replicates to ensure reproducibility. Specific statistical tests used for each experiment, including the types of post-hoc tests and significance levels, are described in the corresponding figure legends.

### Reporting summary
Further information on research design is available in the Nature Portfolio Reporting Summary linked to this article.

## Data availability
A reporting summary for this article is available as a Supplementary Information file. The data supporting the findings of this study are available within the article and its supplementary figures. Source data are provided with this paper.

## Code availability
The EMMP pipeline source code is available on CodeOcean (https://codeocean.com/capsule/8527428/tree/v1).

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

## Acknowledgements

The authors thank Dr. Alan Bear and Nirjal Bhattarai for help with the AAV manufacturing protocol. This project was supported by the Center for Biologics Evaluation and Research, US Food and Drug Administration, and by an appointment to the ORISE Research Participation Program at the CBER, U.S. Food and Drug Administration, administered by the Oak Ridge Institute for Science and Education through an interagency agreement between the U.S. Department of Energy and FDA/Center. We thank Dr. Zhaohui Ye, Dr. Jianyang Wang, and Dr. Ha-Na Lee for reviewing and providing helpful comments on the manuscript.

## Author contributions

The conceptualization of the study was led by S.B., A.M.S., and R.M., who developed the research framework and objectives. A.E., S.S. (Sean Smith), and L.S.Q. developed the EMMP. S.B. developed the validation method. S.B., A.M.S., S.W., S.S. (Sima Saleh), R.J.D., and S.N. contributed to the methodology, designing, and implementing the experimental approaches. R.M. provided supervision, ensuring the study's direction and scientific rigor. The original draft of the manuscript was written by S.B. and R.M. All authors reviewed and approved the final manuscript.

## Competing interests

R.J.D. reports a relationship with Waters Corporation that includes employment. The remaining authors declare no competing interests.

 
