## [Transparent Peer Review file · Nature Communications]

Integrated computational and experimental immunoengineering of Adeno-associated virus capsid T cell epitopes in mice

Corresponding Author: Dr Ronit Mazor

Version 1:

Reviewer comments:

Reviewer #1

(Remarks to the Author)

The manuscript by Bing et al. describes an Epitope Modification and MHC Prediction (EMMP) tool, which evaluates amino acid substitutions for their predicted effects on major histocompatibility complex (MHC) binding with the goal to render AAV vectors non immunogenic to reduce rejection of transduced cells and to prevent induction of vector neutralizing antibodies that could interfere with redosing.

To this end they identify an immunodominant MHC class II binding epitope in BALB/c mice and claim that figure 2D and E show that this epitope is recognized by CD4 but not CD8 T cells; the data are not convincing, and statistical differences (if there are any) are not shown for these 2 figures. Based on their prediction algorithm they selected 26 mutants for further analyses. Two of the mutant AAVs were able to transduce cells in vitro and showed similar biodistribution in mice to wild-type AAV although one of the mutants was less efficacious. In Figure 6 the authors show the less effective mutant failed to elicit both mutants fail to elicit T cell responses while the response to the other mutant was attenuated. Responses for the CD4 subset are not convincing (Fig. 6 C-F) as responses to the wildtype virus are negligible. Binding antibody responses are only marginally reduced. Sera were not tested for AAV neutralizing antibodies.

The results are unlikely to have relevance to human gene therapy. HLA is by far more diverse than mouse MHC and while mice commonly respond to a single epitope, human T cells tend to recognize multiple epitopes on viral proteins. Considering that the authors had to screen 23 vectors for a single epitope the number of possible mutations in a protein with multiple epitopes would be staggering. Also it is far from certain that viral capsid with multiple mutations would retain the ability to grow or to transduce cells. On top of this for each patient vectors would have to be custom made, which is not feasible. In addition, the effects were marginal on antibody responses. Effects of the mutations on T cells are hard to evaluate as responses to the wildtype virus were too low. A more immunogenic platform to present the AAV capsid protein (such as by an adenovirus vector) may overcome this problem.

(Remarks on code availability)

When I tried to access the EMMP tool I got this: An EMMP (Environmental Monitoring and Management Plan) is a crucial tool for proactively managing and monitoring the environmental impacts of a project, particularly in construction or development activities

Reviewer #2

(Remarks to the Author)

In this manuscript, Bing et al. present a combined experimental and computational workflow designed to identify mutations that reduce anti-AAV9 antibody titres and so increase the therapeutic efficacy of AAV9-based drugs.

The approach relies first on experimental assessment to determine whether the anti-drug response is CD4-helper T cell assisted or otherwise. In tandem, the authors apply a pooled peptide stimulation assay to identify which fragments induce higher rates of IL2-secretion, implying they activate T cells via presentation on host MHC. A computational prediction

workflow then helps to narrow down on the most likely peptide epitope, and which mutations to the epitope might be maximally deleterious to its ability to be presented on cell surfaces. Validation of this approach in a mouse model revealed a mutant (R312Q) that significantly reduced H2-IEd presentation of an AAV9-component peptide, reflected by a marked reduction in T cell activation biomarkers and net anti-AAV antibody titre, at the expense of moderately compromised gene delivery fitness.

This proof of principle is an interesting read, and the claims appear well supported by experimental results. Overall, the paper reports a successful pipeline application of peptide stimulation followed by application of NetMHCII2.3 to home in on a subset of mutants for experimental testing, where relevant murine MHC alleles of interest are known a priori.

Notwithstanding its utility in this case study, the EMMP package released with the paper is essentially a wrapper that iterates over all potential single-point substitutions, insertions, and deletions, and feeds them into existing MHC-presentation prediction algorithms; EMMP is better viewed as a novel pipeline rather than a novel prediction tool.

Major Comments

- This study appears methodologically very similar to Bing et al. Nature Biomedical Engineering (2023): 8,193–220; the major distinction appears to be the mutant generation strategy: rather than using an AAV9/AAV5 chimera approach, here the authors perform exhaustive single-point mutagenesis (including insertions/deletions). The relationship between the former work and this paper should be better characterised in the introduction, so it is clearer to determine the areas of novelty contributed by this study and the associated impact.
- The wording in favour of EMMP is quite strong, for example, page 8 line 256: “The EMMP tool offers a significant advancement over traditional methods for MHC-binding prediction”. EMMP streamlines the application of NetMHCII2.3, but it does not offer advancement in the task of MHC-binding prediction. Statements made throughout the text ought to be adjusted to more accurately describe EMMP’s impact.
- BALB/c mice have two defined MHC class II alleles (H2-IEa and H2-IEd), meaning it was tractable to use NetMHCII2.3 with these defined alleles. Applied to humans, whose MHC alleles are much more heterogeneous, it would be considerably more practical to use NetMHCIIpan4.0 (or NetMHCpan4.1 for class I). As it pertains to the potential extensibility of the approach to human data, could the authors confirm analogous results are obtained using NetMHCIIpan4.0 for epitope deduction/mutant stratification?
- Page 4, line 109-110: The authors refer to the “1.3 percentile”, but it’s not clear what this is in reference to? It implies a benchmark set of known binders to this allele with measuring affinities.
- Table 1: (a) The AAV9_VI epitope, ranked in the 40th percentile, appears to be in a different 15-mer sliding window to the wildtype AAV9; AAV9_VI epitope starts ‘KRLN’, which is 9 residues into the reported wildtype AAV9 epitope. Could the authors report the percentile of the positionally equivalent 15-mer for AAV9_VI (starting ‘NNN’)? I suspect this will be much closer in percentile to the wildtype. (b) the percentile values for all mutants should be recorded somewhere (e.g. an SI table) for completeness. (c) the table would be easier to digest if ordered within mutation type by residue position, or by percentile binding probability. (d) it’s unclear why the AAV9 wildtype and mutant epitopes are presented as 17-mers.
- Figure 1A: Could the authors clarify if the 18 mice had separate assays performed on them (3 times) or if the cells from the 18 mice were pooled and then an assay was run on them (3 times). If it is the latter, ANOVA may not be appropriate to assess the effects of each group due to the low sample size. A non-parametric version of the test might be more appropriate.
- Figure 2A: ANOVA is used to assess differences between groups, but each group only has 3 samples within. It might be more appropriate to use a non-parametric version of ANOVA for the statistical tests.
- Figure 6: This figure confusing and difficult to interpret.
 - o Fold-change is used, but it is not clear what it is relative to.
 - o The X-axis and colour bars share the same labels, but what is the difference between them?
 - o What is “99+106”?
 - o The title is “AAV9 mutant do not activate the CD4 T cells to produce IFN-γ and IL-2” but would read better as “The R312Q AAV9 mutant does not activate the CD4 T cells to produce IFN-γ and IL-2”.

Minor Comments

- Could the authors discuss the limitations of using a computational tool that is designed purely for predicting peptide presentation rather than direct prediction of peptide immunogenicity?
- The authors should be precise which algorithms their software calls (‘IEDB MHC I or MHC binding tools’ is too general)
- Figure 2A: There are two colours used in the bars that seem to correspond to different y-axis with different scales for IL-2 versus IFNγ. Could these be separated into two plots, one for IL-2 and one for IFNγ, to improve readability?
- Figure 3B: Colour bars should have a label (with units if appropriate)
- Figure 4: The description for panel B is listed as C in the description text. This should be changed.
- Figure 5B: The error bars are denoted by SD here, but in the rest of the text, they are SEM. This should be standardised throughout the work.

(Remarks on code availability)

- Could a table like Table 1 in the text be output by EMMP? The current implementation requires the users to do further analysis to come up with a list of the most beneficial changes to their input peptide based on the three different tracks.
- Like the recommendation in the main text, the heatmaps output by EMMP should have labels describing their colour bars.
- The main python module uses comments to describe what a function or method does. These should be converted to

docstrings.

- The code uses a mix of CamelCase and snake_case to name functions, variables, classes, etc. For readability purposes, I recommend standardising this throughout the code so that all variable names follow the same convention, all functions follow the same convention, etc.
- The main classes could make use of attributes and static methods for properties of the class that do not depend on the specific instance of the object. There are lots of times where “self” is passed into a method and not used. These parts could be refactored to follow better object-oriented coding principles.

Reviewer #3

(Remarks to the Author)

(Remarks on code availability)

Version 2:

Reviewer comments:

Reviewer #1

(Remarks to the Author)

The authors have addressed most of my queries. The main problem - that epitope modifications are unlikely to have clinical relevance could as expected not be addressed as this is inherent to the theme of the manuscript. Nevertheless, the new software may be of value for preclinical experiments.

(Remarks on code availability)

Reviewer #2

(Remarks to the Author)

We thank the authors for their point-by-point response to all of our previous comments.

-We note that updating the MHC presentation model to the latest generation (i.e. NetMHCIIpan4) leads to prediction of both wildtype and mutant peptides as “presentable”, meaning the same mutants would not have been highlighted for downstream testing. However, we equally appreciate that it is not always the case that more recent deep learning models that have higher headline accuracy achieve this via improved understanding of the underlying mechanisms of the biology, especially in high noise/incomplete data settings.

While we appreciate the inclusion of a discussion point relating to this, I think the Table presented in response to comment #3 should be included in the Supporting Information of the paper to appropriately convey the differences of conclusion reached by different NetMHCII models.

As only one case study is examined here, it is unclear the extent to which we might expect the interpretations offered by NetMHCII2.3 specifically to be the most accurate in other cases. This may affect the perceived impact of the work.

-We appreciate the reply and manuscript change implemented in response to our minor comment #8, but request the authors thoroughly review all other parts of the manuscript to ensure they are not claiming that NetMHC can predict immunogenicity when it can only predict peptide presentability. E.g. the textual addition in response to Major comment #3 “While NetMHCII2.3 predicted R312H and R312Q as having reduced immunogenicity” should strictly be “While NetMHCII2.3 predicted R312H and R312Q as having reduced presentability”, etc.

-The edits to Figure 6 have improved the figure, but a label should be added to the colour bars for ease of interpretation.

-We appreciate the authors splitting Figure 2A into two separate figures. Please could they confirm that Dunnett’s test was also performed with a non-parametric version of a t-test? Also, now that E and F have been added, would a non-parametric test be more appropriate there as well?

-As a final observation (which we only realised upon re-reviewing the manuscript): it may be appropriate to assess new mutants in all new potential sliding windows of the protein sequence that contain that mutation, to ensure that a new presentable epitope has not inadvertently been created in a different N-mer frame. This check could optionally be introduced as a natural extension of the current EMMP algorithm.

(Remarks on code availability)

The authors addressed our comments about the codebase in their point-by-point responses, and implemented changes that addressed our previous concerns. Our final point in this review could optionally be implemented as an extension to the current workflow.

Reviewer #3

(Remarks to the Author)

(Remarks on code availability)

I co-reviewed this manuscript with one of the reviewers who provided the listed reports. This is part of the Nature Communications initiative to facilitate training in peer review and to provide appropriate recognition for Early Career Researchers who co-review manuscripts

Version 3:

Reviewer comments:

Reviewer #2

(Remarks to the Author)

We thank the authors for further revising their manuscript. We have no further points to raise.

(Remarks on code availability)

Reviewer #3

(Remarks to the Author)

(Remarks on code availability)

Point-by-point response to Comments from the reviewers and Editor:

We thank the reviewers for the favorable evaluation and the helpful comments. We have gone over the comments and carefully revised the manuscript, as explained in the point-by-point response, below. Reviewer comments have been included in their entirety and are in black text. Responses are in blue text.

<Reviewer #1>

Remarks to the Author

The manuscript by Bing et al. describes an Epitope Modification and MHC Prediction (EMMP) tool, which evaluates amino acid substitutions for their predicted effects on major histocompatibility complex (MHC) binding with the goal to render AAV vectors non immunogenic to reduce rejection of transduced cells and to prevent induction of vector neutralizing antibodies that could interfere with redosing.

To this end they identify an immunodominant MHC class II binding epitope in BALB/c mice and claim that figure 2D and E show that this epitope is recognized by CD4 but not CD8 T cells; the data are bon convincing, and statistical differences (if there are any) are not shown for these 2 figures.

Response: We thank the reviewer for this important observation. We have addressed this concern by updating the statistical analysis and figure presentation. The missing statistical significance indicators in the original Figure 2D and E have now been added to the revised Figure 2E and F. The updated figures now clearly show statistical differences between CD4+ and CD8+ T cell responses using appropriate statistical tests (two-way ANOVA with Dunnett's multiple comparisons test, **** $p < 0.0001$), confirming that the identified epitope is indeed recognized by CD4+ but not CD8+ T cells. We note that the CD8 responses (represented by green legend and placed after the orange bars) was zero which may make it difficult to see in the graph. The comment "statistical differences (if there are any)" raised a suspicion that the reviewer may have mixed the splenocytes bars (orange) with the green bars (CD8) or missed the CD8 bars altogether. To clarify this in the text, the following statement is written in line 111: "while CD8⁺ T cells showed no response in ELISpot assays".

Based on their prediction algorithm they selected 26 mutants for further analyses. Two of the mutant AAVs were able to transduce cells in vitro and showed similar biodistribution in mice to

wild-type AAV although one of the mutants was less efficacious. In Figure 6 the authors show the less effective mutant failed to elicit both mutants fail to elicit T cell responses while the response to the other mutant was attenuated. Responses for the CD4 subset are not convincing (Fig. 6 C-F) as responses to the wildtype virus are negligible.

Response: We acknowledge the reviewer's concern regarding the CD4+ T cell responses in Figure 6C-F. The reviewer correctly notes that the wildtype virus responses, even though statistically significant, appear modest in the flow cytometry data. However, when the same splenocytes from immunized mice were analyzed using different experimental approaches (ELISpot assays in Figure 6A, B versus flow cytometry in Figure 6C-F), the differences between wildtype and mutant responses were much more pronounced and statistically significant in the ELISpot analysis demonstrating 2.5-5 fold increase. This discrepancy can be explained by the different sensitivities and methodological approaches of these two assays. ELISpot assay measures cumulative cytokine secretion over 24 hours, providing high sensitivity for detecting low-frequency responding cells. In contrast, flow cytometry captures a snapshot of intracellular cytokine production at a single time point (after 4-hour GolgiPlug treatment), which may be less sensitive for detecting subtle differences. The relatively low baseline responses to wildtype virus in the flow cytometry data reflect the physiological reality that T cell responses in this mouse model are modest but still detectable and biologically relevant.

Binding antibody responses are only marginally reduced. Sera were not tested for AAV neutralizing antibodies.

Response: In response to this comment, we have now performed neutralizing antibody assays and added new data in Figure 7C and D. Key findings from the neutralizing antibody analysis show that similar to the binding antibody responses, neutralizing antibody titers against the mutant capsids were reduced compared to wildtype AAV9. R312H showed statistically significant reduction in neutralizing antibody titers ($p < 0.05$), while R312Q also showed reduced neutralizing activity, though the effect was more modest.

We have added the following text to the manuscript:

“To further evaluate the functional significance of reduced binding antibodies, we assessed the neutralizing capacity of sera from immunized mice. Neutralizing antibody analysis revealed that both mutant vectors, particularly R312H, showed significantly reduced neutralizing activity compared to wildtype AAV9 (Figure 7C, D), indicating that epitope modification can reduce functionally relevant immune responses that could impact vector re-administration.”

The results are unlikely to have relevance to human gene therapy. HLA is by far more diverse than mouse MHC and while mice commonly respond to a single epitope, human T cells tend to recognize multiple epitopes on viral proteins. Considering that the authors had to screen 23 vectors for a single epitope the number of possible mutations in a protein with multiple epitopes would be staggering. Also it is far from certain that viral capsid with multiple mutations would retain the ability to grow or to transduce cells. On top of this for each patient vectors would have to be custom made, which is not feasible. In addition, the effects were marginal on antibody responses. Effects of the mutations on T cells are hard to evaluate as responses to the wildtype virus were too low. A more immunogenic platform to present the AAV capsid protein (such as by an adenovirus vector) may overcome this problem.

Response: We appreciate the reviewer's thoughtful comments regarding the translational relevance of our approach. We agree that human HLA diversity and the multi-epitope nature of T-cell responses present major challenges compared to the single-epitope response often observed in mice. To address this, we used an allele panel designed for broad human population coverage, as described in the IEDB reference sets for HLA alleles and population coverage tools¹. EMMP automatically evaluates each amino acid substitution across 27 representative HLA alleles and reports median, minimum, and maximum predicted MHC-binding values, providing a concise summary of how consistently a substitution may reduce binding across diverse genotypes.

We acknowledge that extensive multi-epitope editing could compromise AAV capsid assembly and transduction. Therefore, EMMP is intended as a prioritization framework, not a comprehensive editing solution, to identify a small set of substitutions that balance reduced immunogenicity with structural feasibility.

In response to the reviewer's suggestion regarding more immunogenic presentation systems, we accepted this challenge. We note that the adenoviral hexon is known as the major immunogenic determinant of adenovirus^{2, 3}, and several hexon mutations that reduce immunogenicity have been patented⁴. To validate EMMP's broader applicability, we applied it to three reported HLA-A02:01-restricted CD8⁺ T cell epitopes in hexon protein (GLVDCYINL, LLYANSAHA, YVLFVFDV). EMMP analysis successfully identified substitutions predicted to reduce MHC binding (colored in blue below), with results for each epitope shown below in sequential order for both HLA-A02:01-specific binding (left panels) and minimum binding values across all 27 representative alleles (right panels). Notably, several computationally predicted mutations corresponded to empirically validated modifications described in the granted patent (L520P, A900S, V925K, highlighted in green boxes), demonstrating that EMMP can identify immunomodulatory substitutions consistent with published and patented data, supporting its general applicability to other viral capsid proteins. This cross-validation supports the general applicability of EMMP to other viral capsid proteins beyond AAV.

[Figure Redacted]

Figure 1. Removal of CD8 T cell epitopes in adenoviral hexon. ICOVIR15K is an oncolytic virus designed to replicate in and lyse cancer cells. ICOVIR15K-QD has been engineered by Bazan et al, to reduce immunogenicity by mutations L520 and V925K. The figures above were taken from ⁴. (A) Binding affinity of mutated epitopes in ICOVIR15K-TD-gp100-tyr compared to wild-type epitopes in HA5 showing significantly reduced LA-A2 binding by almost 600 times compared to its unmodified counterpart. B. Immune responses generated after intratumoral administration of 10^{10} vp/tumor of ICOVIR15K or ICOVIR15K-QD in transgenic HLA-A2.1 mice. ELISPOT was performed using 2,500,000 splenocytes per well. B16+IFN refers to reactivity against B16CAR-A2 tumors previously incubated with IFN- γ . This figure demonstrates a dramatic reduction in CD8 immunogenicity against the adenoviral hexon In vivo using mutations that were predicted un EMMP.

We have added the following text to the manuscript:

“While we acknowledge the challenges of translating findings from a single mouse epitope to the complex multi-epitope landscape of human immune responses, our EMMP approach is designed to address HLA diversity systematically. By evaluating mutations across 27 representative HLA alleles that provide maximal population coverage, EMMP can identify substitutions that consistently reduce immunogenicity across diverse genetic backgrounds. The broader applicability of EMMP is demonstrated by its successful application to adenoviral hexon epitopes, where computationally predicted mutations corresponded to empirically validated modifications that have been patented for clinical use (data not shown). This suggests that EMMP can serve as a general framework for rational deimmunization across multiple viral vector platforms.”

Remarks on code availability

When I tried to access the EMMP tool I got this: An EMMP (Environmental Monitoring and Management Plan) is a crucial tool for proactively managing and monitoring the environmental impacts of a project, particularly in construction or development activities

Response: we assume that the reviewer refers to googling the acronym EMMP. Considering that the manuscript is still under review, there should be no expectation to find the Epitope Modification and MHC Prediction (EMMP) online. The EMMP (Epitope Modification and MHC Prediction) pipeline source code is now available on CodeOcean.

References

1. Greenbaum J, Sidney J, Chung J, Brander C, Peters B, Sette A. Functional classification of class II human leukocyte antigen (HLA) molecules reveals seven different supertypes and a surprising degree of repertoire sharing across supertypes. *Immunogenetics* **63**, 325-335 (2011).
2. Sumida SM, *et al.* Neutralizing antibodies to adenovirus serotype 5 vaccine vectors are directed primarily against the adenovirus hexon protein. *J Immunol* **174**, 7179-7185 (2005).
3. Tang J, *et al.* Human CD8+ cytotoxic T cell responses to adenovirus capsid proteins. *Virology* **350**, 312-322 (2006).
4. BAZAN RAG. Oncolytic adenoviruses with mutations in immunodominant adenovirus epitopes and their use in cancer treatment.

<Reviewer #2>

Remarks to the Author

In this manuscript, Bing et al. present a combined experimental and computational workflow designed to identify mutations that reduce anti-AAV9 antibody titers and so increase the therapeutic efficacy of AAV9-based drugs.

The approach relies first on experimental assessment to determine whether the anti-drug response is CD4-helper T cell assisted or otherwise. In tandem, the authors apply a pooled peptide stimulation assay to identify which fragments induce higher rates of IL2-secretion, implying they activate T cells via presentation on host MHC. A computational prediction workflow then helps to narrow down on the most likely peptide epitope, and which mutations to the epitope might be maximally deleterious to its ability to be presented on cell surfaces. Validation of this approach in a mouse model revealed a mutant (R312Q) that significantly reduced H2-IE d presentation of an AAV9-component peptide, reflected by a marked reduction in T cell activation biomarkers and net anti-AAV antibody titer, at the expense of moderately compromised gene delivery fitness.

This proof of principle is an interesting read, and the claims appear well supported by experimental results. Overall, the paper reports a successful pipeline application of peptide stimulation followed by application of NetMHCII2.3 to home in on a subset of mutants for experimental testing, where relevant murine MHC alleles of interest are known a priori.

Notwithstanding its utility in this case study, the EMMP package released with the paper is essentially a wrapper that iterates over all potential single-point substitutions, insertions, and deletions, and feeds them into existing MHC-presentation prediction algorithms; EMMP is better viewed as a novel pipeline rather than a novel prediction tool.

Response: We thank the reviewer for this thoughtful assessment and agree that EMMP should be characterized as a novel computational pipeline rather than a novel prediction algorithm. We appreciate this clarification and revise the manuscript accordingly to avoid the use of words such as prediction tool or method. These words were replaced with pipeline when appropriate.

The reviewer correctly identifies that EMMP serves as a systematic wrapper that automates the application of existing MHC-binding prediction tools across comprehensive mutation landscapes. However, we believe this automation represents a significant practical advancement for the field. Prior to EMMP, researchers would need to manually evaluate hundreds or thousands of potential mutations individually, which is both time-consuming and

error prone. EMMP transforms this intractable manual process into an automated, systematic analysis that can be completed in minutes rather than weeks.

Furthermore, EMMP's ability to simultaneously analyze mutations across multiple HLA alleles and provide summary reports (median, minimum, maximum binding values) addresses a critical gap in translating computational predictions to clinical applications, where population-level HLA diversity must be considered.

Major Comments

1. This study appears methodologically very similar to Bing et al. Nature Biomedical Engineering (2023): 8,193–220; the major distinction appears to be the mutant generation strategy: rather than using an AAV9/AAV5 chimera approach, here the authors perform exhaustive single-point mutagenesis (including insertions/deletions). The relationship between the former work and this paper should be better characterised in the introduction, so it is clearer to determine the areas of novelty contributed by this study and the associated impact.

Response: We thank the reviewer for this important observation. We acknowledge that our current study builds upon our previous work (Bing et al., Nature Biomedical Engineering 2024, 8:193-200), and we agree that the relationship between these studies should be better clarified. The following content has been added to the Introduction section (line 56-67):

“Previously, we demonstrated that rational epitope modification through chimeric design could successfully eliminate immunodominant T-cell responses to AAV9 in human PBMCs. In that study, multiple amino acids within the immunodominant epitope were replaced with corresponding sequences from AAV5, resulting in complete elimination of T-cell activation. However, the chimeric approach was limited to cases where suitable non-immunogenic sequences existed in related viral serotypes. Building upon this foundation, the current study addresses the need for a more systematic and broadly applicable approach to epitope modification. Here, we developed the Epitope Modification and MHC Prediction (EMMP) pipeline to comprehensively evaluate all possible single-point mutations, insertions, and deletions within immunogenic epitopes, enabling rational design even when suitable chimeric sequences are not available. Furthermore, we validate this approach in vivo using a mouse model and demonstrate its broader applicability to other viral vectors.”

2. The wording in favour of EMMP is quite strong, for example, page 8 line 256: “The EMMP tool offers a significant advancement over traditional methods for MHC-binding prediction”. EMMP streamlines the application of NetMHCII2.3, but it does not offer advancement in the task of MHC-binding prediction. Statements made throughout the text ought to be adjusted to more accurately describe EMMP's impact.

Response: We thank the reviewer for this important clarification. We agree that our language should more accurately reflect EMMP's contribution as a systematic pipeline rather than claiming advancement in MHC-binding prediction algorithms themselves. EMMP leverages existing, well-established prediction tools and provides value through automation and systematic analysis rather than improving the underlying prediction accuracy. We have revised the manuscript to more precisely describe EMMP's impact as streamlining and automating the

application of existing MHC-binding prediction methods, rather than advancing the prediction algorithms themselves.

“The EMMP pipeline offers a significant advancement in the systematic application of MHC-binding prediction methods by automating comprehensive mutation analysis.” (line 275-276).

Similar word replacement and edits were done in the abstract (line 26), the introduction (lines 80-83), the results (lines 123, 127 and 135), discussion (lines 269 and 279) and methods (line 444 and 449)

3. BALB/c mice have two defined MHC class II alleles (H2-IEa and H2-IEd), meaning it was tractable to use NetMHCII2.3 with these defined alleles. Applied to humans, whose MHC alleles are much more heterogeneous, it would be considerably more practical to use NetMHCIIpan4.0 (or NetMHCPan4.1 for class I). As it pertains to the potential extensibility of the approach to human data, could the authors confirm analogous results are obtained using NetMHCIIpan4.0 for epitope deduction/mutant stratification?

Response: Our selection of NetMHCII2.3 was based on it being the most up-to-date and IEDB-recommended method available when our experiments were conducted in 2021-2022. More recently, NetMHCIIpan4.0 EL has become the newly recommended approach. Therefore, we provide here a comparative analysis across six different prediction algorithms available in IEDB to assess the consistency of predictions.

peptide	consensus	NetMHCII1.1 (SMM-align)	NetMHCII2.3 (NN-align 2.3)	NetMHCIIpan4.3 EL	NetMHCIIpan4.2 EL	NetMHCIIpan4.1 EL
NWGFRPKRLNFKLFN WT	6.8	12	1.6	0.83	0.82	0.7
NWGFRPKHLNFKLFN R312H	16	19	13	0.52	0.55	0.56
NWGFRPKQLNFKLFN R312Q	22.5	33	12	0.83	0.75	0.77

Comparison across prediction methods reveals significant variability. Notably, different algorithms produced markedly divergent results. The SMM-align method failed to predict even the wild-type sequence as immunogenic (colored in red) assuming a cut point of 10% for immunogenicity potential, despite our experimental validation confirming its strong immunogenicity. In contrast, NetMHCII2.3 predicted our mutations as less immunogenic (higher percentile ranks), whereas NetMHCIIpan4.0 EL predicted them as still highly immunogenic (lower percentile ranks). This analysis highlights that the value of our EMMP pipeline lies in its systematic automation rather than enhanced prediction accuracy, and underscores that experimental validation remains indispensable for confirming reductions in immunogenicity. The following content has been added to the discussion section (line 302-312):

“An important limitation of our computational approach is the dependence on the choice of MHC-binding prediction algorithm. Our study utilized NetMHCII2.3, which was the most current and IEDB-recommended method at the time of our experimental work (2021–2022). However, comparison with the more recently recommended NetMHCIIpan4.0 EL method reveals substantially different predictions for our selected mutations. While NetMHCII2.3 predicted R312H and R312Q as having reduced immunogenicity, NetMHCIIpan4.0 EL suggests these mutations would remain highly immunogenic. This discrepancy highlights the evolving nature of prediction algorithms and underscores the importance of experimental validation over computational predictions alone. Future applications of the EMMP pipeline should incorporate

multiple prediction methods and prioritize empirical validation to ensure robust candidate selection across different algorithmic approaches”

4. Page 4, line 109-110: The authors refer to the “1.3 percentile”, but it’s not clear what this is in reference to? It implies a benchmark set of known binders to this allele with measuring affinities.

Response: We thank the reviewer for pointing out this lack of clarity. To determine the precise binding core within the immunogenic region, we analyzed overlapping sequences between peptides 102 and 103, which both showed positive responses in our experiments. Using IEDB NetMHCII2.3 prediction, we evaluated the binding affinity of these overlapping sequences to H2-IAd and H2-IEd (new Supplementary Table 1). Among the tested sequences, NNWGFRPKRLNFKLF was predicted to have the strongest binding with the lowest percentile rank of 1.2.

We have corrected two errors in the original manuscript: (1) the percentile rank should be 1.2, not 1.3 as originally stated, and (2) the sequence should be NNWGFRPKRLNFKLF, not NNNWGFRPKRLNFKL (a shift of one amino acid). These corrections have been made in the revised manuscript and are reflected in the new Supplementary Table 1.

Supplementary Table 1. In silico analysis of identified epitope

Peptide #	Sequence	Percentile rank	
		H2-IAd	H2-IEd
102	NNWGFRPKRLNFKLF	86	1.2
	NWGFRPKRLNFKLFN	84	1.6
	WGFRPKRLNFKLFNI	82	2.7
103	GFRPKRLNFKLFNIQ	77	13

Binding affinity prediction using NetMHCII2.3 method, which was the most up-to-date version available at the time of analysis

Table 1: (a) The AAV9_VI epitope, ranked in the 40th percentile, appears to be in a different 15-mer sliding window to the wildtype AAV9; AAV9_VI epitope starts ‘KRLN’, which is 9 residues into the reported wildtype AAV9 epitope. Could the authors report the percentile of the positionally equivalent 15-mer for AAV9_VI (starting ‘NNN’)? I suspect this will be much closer in percentile to the wildtype. (b) the percentile values for all mutants should be recorded somewhere (e.g. an SI table) for completeness. (c) the table would be easier to digest if ordered within mutation type by residue position, or by percentile binding probability. (d) it’s unclear why the AAV9 wildtype and mutant epitopes are presented as 17-mers.

Response: We thank the reviewer for these detailed suggestions to improve Table 1 clarity and completeness.

(a) AAV9_VI epitope comparison: The AAV9VI sequence (KRLNVKIFNIQVKEV) represents a human epitope with a different binding core compared to the mouse epitope binding core (GFRPKRLNV) identified in our study. When we predict the positionally equivalent 15-mer for AAV9_VI starting with the same position as our mouse epitope (NNWGFRPKRLNVKIF), the percentile rank is 1.7. However, since this represents a human epitope with a different binding core, direct comparison with our mouse epitope data would not be meaningful. Therefore, we have removed AAV9_VI from the revised table to avoid confusion.

(b) Complete percentile values: All percentile ranks for the mutations are displayed in the original Figure 3B, which shows the comprehensive EMMP analysis results. We have updated the color scheme in Figure 3B to better visualize low binding spots and facilitate interpretation of the results.

(c) Table organization: We have reorganized the table as suggested, ordering mutations by residue position and then by percentile rank within each position to improve readability.

(d) Sequence length correction: The sequences were incorrectly described as 17-mers in the original submission. This has been corrected in the revised table. Additionally, we have updated the wild-type sequence to the correct sequence (NNWGF~~R~~PKRLNFKLF).

Table 1. In silico analysis of mutated AAV9

Mutation	AA sequence	Mutation type	Percentile
Wild type	NNWGF ^R PKRLNFKLF		1.2
F308D	NNWGD ^R RPKRLNFKLF	Replacement	29
F308G	NNWGG ^R RPKRLNFKLF	Replacement	26
F308N	NNWGN ^R RPKRLNFKLF	Replacement	23
F308P	NNWGP ^R RPKRLNFKLF	Replacement	17
F308E	NNWGE ^R RPKRLNFKLF	Replacement	16
F308S	NNWGS ^R RPKRLNFKLF	Replacement	16
F308Q	NNWGQ ^R RPKRLNFKLF	Replacement	14
F308T	NNWGT ^R RPKRLNFKLF	Replacement	9.9
R309D	NNWGF ^D PKRLNFKLF	Replacement	23
R309G	NNWGF ^G PKRLNFKLF	Replacement	22
R309N	NNWGF ^N PKRLNFKLF	Replacement	14
R309P	NNWGF ^P PKRLNFKLF	Replacement	14
R309S	NNWGF ^S PKRLNFKLF	Replacement	11
R309T	NNWGF ^T PKRLNFKLF	Replacement	8.9
R312D	NNWGF ^R PK ^D LNFKLF	Replacement	18
R312N	NNWGF ^R PK ^N LNFKLF	Replacement	15
R312G	NNWGF ^R PK ^G LNFKLF	Replacement	11
R312H	NNWGF ^R PK ^H LNFKLF	Replacement	11
R312Q	NNWGF ^R PK ^Q LNFKLF	Replacement	9.6
R312E	NNWGF ^R PK ^E LNFKLF	Replacement	7.7
F308^R309insD	NNWGF ^D RPKRLNFKLF	Insertion	14
F308^R309insG	NNWGF ^G RPKRLNFKLF	Insertion	14
F308^R309insN	NNWGF ^N RPKRLNFKLF	Insertion	9
R309^P310insG	NNWGF ^R GPKRLNFKLF	Insertion	9.9
F308del	NNWG_ ^R PKRLNFKLF	Deletion	19
R309del	NNWGF_ ^P KRLNFKLF	Deletion	12

EMMP predictions were generated using HLA binding data from the IEDB NetMHCII 2.3 prediction method, which was the most up-to-date version available at the time of analysis.

5. Figure 1A: Could the authors clarify if the 18 mice had separate assays performed on them (3 times) or if the cells from the 18 mice were pooled and then an assay was run on them (3 times). If it is the latter, ANOVA may not be appropriate to assess the effects of each group due to the low sample size. A non-parametric version of the test might be more appropriate.

Response: Each of the 18 mice was processed individually, and ELISpot assays were performed in triplicate for each mouse. Thus, we have 18 independent biological replicates, providing sufficient sample size for ANOVA. This experimental design and statistical approach have been clarified in the revised Methods and figure legend sections.

6. Figure 2A: ANOVA is used to assess differences between groups, but each group only has 3 samples within. It might be more appropriate to use a non-parametric version of ANOVA for the statistical tests.

Response: We agree with the reviewer's suggestion. Given the small sample size (n=3) in each group, we have re-analyzed the data using non-parametric statistical tests and updated the statistical methods accordingly. Additionally, following the reviewer's suggestion in the minor

comments to separate IL-2 and IFN- γ graphs, we have divided the original Figure 2A into Figure 2A and 2B for better clarity.

7. Figure 6: This figure confusing and difficult to interpret.

- 1) Fold-change is used, but it is not clear what it is relative to.

Response: The fold-change values represent the ratio of peptide-stimulated responses compared to medium-only control (no peptide) for each respective treatment group. We have clarified this in the revised figure legend.

“Fold changes were calculated by comparing peptide-stimulated responses to medium-only control (no peptide) for each treatment group.”

- 2) The X-axis and colour bars share the same labels, but what is the difference between them?

Response: The X-axis represents the different AAV vectors that were injected into BALB/c mice (AAV9 wild-type, R312H mutant, R312Q mutant, or sham control). The color bars represent the different peptides used to restimulate the splenocytes isolated from these mice. We have updated the X-axis labels in Figure 6 and clarified this distinction in the revised figure legend.

“X-axis indicates the type of AAV vector injected into mice. Color bars indicate the peptides used for restimulation: irrelevant peptide (neg (99+106)), wild-type peptide sequence (WT (102+103)), or mutant peptide sequences (R312H (102+103), R312Q (102+103)).”

- 3) What is “99+106”?

Response: “99+106” refers to a combination of two irrelevant control peptides that were selected as negative controls. During our initial epitope screening, we generated a total of 242 overlapping peptides spanning the entire AAV9 sequence with overlap of 12 amino acids. From this library, we selected peptides 99 and 106 as negative controls because they had low predicted MHC-binding affinity and were located in proximity to the immunogenic peptides 102+103, allowing us to control for non-specific responses in the same region. We have clarified this in the methods section (line 424-428).

“For negative controls in T-cell assays, peptides 99 and 106 were selected from the 242-peptide library based on their low predicted MHC class II binding affinity and proximity to the immunogenic region (peptides 102+103). These peptides serve as irrelevant controls to account for non-specific T-cell activation in the same sequence region.”

- 4) The title is “AAV9 mutant do not activate the CD4 T cells to produce IFN- γ and IL-2” but would read better as “The R312Q AAV9 mutant does not activate the CD4 T cells to produce IFN- γ and IL-2”.

Response: We agree with the reviewer's suggestion and have updated the figure title accordingly.

Minor Comments

8. Could the authors discuss the limitations of using a computational tool that is designed purely for predicting peptide presentation rather than direct prediction of peptide immunogenicity?

Response: We thank the reviewer for this important point. We have already addressed this limitation in our Discussion section and have now expanded this discussion to provide more comprehensive coverage of the computational prediction limitations (line 312-325).

“Prediction accuracy may also decrease for novel peptide types or less-studied MHC alleles due to gaps in training data. Importantly, EMMP and the algorithms it utilizes are designed to predict peptide-MHC binding rather than actual immunogenicity. Strong predicted binding is necessary for T-cell recognition but not sufficient to guarantee an immune response. Actual immunogenicity depends on multiple additional factors, including antigen processing and presentation efficiency, T-cell receptor repertoire diversity, central and peripheral tolerance mechanisms, peptide abundance, co-stimulatory signals, and the broader immunological context. Consequently, peptides with moderate predicted binding may still elicit immune responses, while others with strong predicted binding may not. Training data biases can further skew predictions if some MHC alleles are underrepresented, limiting the reliability of predictions across diverse genetic backgrounds. This distinction highlights that EMMP serves as a screening tool to prioritize candidate mutations but cannot definitively predict immunogenicity reduction, underscoring the necessity of experimental validation.”

9. The authors should be precise which algorithms their software calls ('IEDB MHC I or MHC binding tools' is too general)

Response: We have updated the manuscript to specify the exact algorithms used by EMMP. For this study, we used NetMHCII2.3, which was the most up-to-date and IEDB-recommended method available at the time of our analysis (2021-2022). The current version of EMMP has been updated to use NetMHCIIpan4.1, which is now the latest version recommended by IEDB. Specifically, the current EMMP implementation uses default parameters with NetMHCpan 4.1 EL for MHC class I predictions and NetMHCIIpan 4.1 EL for MHC class II predictions. MHC source selections are set to 'human' for human peptides and 'mouse' for mouse peptides, with allele(s) and peptide(s) selected by the user within the Jupyter Notebook interface. We have revised the Methods section and main text (line 129-132) to clearly state these specific algorithms rather than using the general term "IEDB MHC binding tools."

“For this study, EMMP utilized NetMHCII2.3 for MHC class II binding predictions, which was the most current and IEDB-recommended method available at the time of analysis. The current version of EMMP has been updated to use NetMHCIIpan4.1, which is now the latest version recommended by IEDB.”

10. Figure 2A: There are two colours used in the bars that seem to correspond to different y-axis with different scales for IL-2 versus IFN γ . Could these be separated into two plots, one for IL-2 and one for IFN γ , to improve readability?

Response: We thank the reviewer for this suggestion to improve figure clarity. As mentioned in our response to an earlier comment (#6), we have separated the original Figure 2A into two distinct plots: Figure 2A (IL-2 responses) and Figure 2B (IFN- γ responses).

11. Figure 3B: Colour bars should have a label (with units if appropriate)

Response: We have added "Percentile Rank" as the label for color bars in Figure 3B.

12. Figure 4: The description for panel B is listed as C in the description text. This should be changed.

Response: We thank the reviewer for catching this labeling error. We have corrected the figure legend to properly reference panel B.

13. Figure 5B: The error bars are denoted by SD here, but in the rest of the text, they are SEM. This should be standardised throughout the work.

Response: We thank the reviewer for this observation. The use of SD in Figure 5B is correct for this specific experiment. In this study, SD was used for figures presenting representative results from a single experimental run, whereas SEM was used when data from multiple independent experiments were combined.

Remarks on code availability

14. Could a table like Table 1 in the text be output by EMMP? The current implementation requires the users to do further analysis to come up with a list of the most beneficial changes to their input peptide based on the three different tracks.

Response: Code for table generation in exportable CSV format has been added. The output provides a list of all possible replacements, insertions and deletions based on the prediction of one or multiple MHC alleles above a given percentile rank. A representative table is shown below for peptide VPQYGYLTL with MHC H-2-Db and a preantral rank >10% (>10% is the default but can be changed).

Mutation	AA sequence	Mutation type	Percentile
Wildtype	VPQYGYLTL		4.2
V100D	DPQYGYLTL	Replacement	14
V100E	EPQYGYLTL	Replacement	11
V100P	PPQYGYLTL	Replacement	29
T107C	VPQYGYLCL	Replacement	11
T107D	VPQYGYLDL	Replacement	15
T107G	VPQYGYLGL	Replacement	14
T107W	VPQYGYLWL	Replacement	11
L108A	VPQYGYLTA	Replacement	21
L108C	VPQYGYLTC	Replacement	25
L108D	VPQYGYLTD	Replacement	73
L108E	VPQYGYLTE	Replacement	41
L108G	VPQYGYLTG	Replacement	54
L108H	VPQYGYLTH	Replacement	31
L108K	VPQYGYLTK	Replacement	35
L108N	VPQYGYLTN	Replacement	66
L108P	VPQYGYLTP	Replacement	21
L108Q	VPQYGYLTQ	Replacement	46
L108R	VPQYGYLTR	Replacement	33
L108S	VPQYGYLTS	Replacement	41
L108T	VPQYGYLTT	Replacement	33
L108V	VPQYGYLTV	Replacement	10
L108W	VPQYGYLTW	Replacement	13
L108Y	VPQYGYLTY	Replacement	19
P101^Q102insC	VPCQYGYLTL	Insertion	11
P101^Q102insE	VPEQYGYLTL	Insertion	11
Q102^Y103insC	VPQCQYGYLTL	Insertion	12
V100del	_PQYGYLTL	Deletion	14
G104del	VPQY_YLTL	Deletion	12
Y105del	VPQYG_LTL	Deletion	12
L106del	VPQYGY_TL	Deletion	11
T107del	VPQYGYL_L	Deletion	18
L108del	VPQYGYLT_	Deletion	68

15. Like the recommendation in the main text, the heatmaps output by EMMP should have labels describing their colour bars.

Response: Code has been added to display “Percentile rank” under the color bar.

16. The main python module uses comments to describe what a function or method does. These should be converted to docstrings.

Response: Comments describing functions/methods have been converted to docstrings.

17. The code uses a mix of CamelCase and snake_case to name functions, variables, classes, etc. For readability purposes, I recommend standardising this throughout the code so that all variable names follow the same convention, all functions follow the same convention, etc.

Response: Modified the code to change methods to snake_case and classes to camelCase based on the PEP 8 – Style Guide for Python Code ([peps.python.org](https://peps.python.org/pep-0008/)).

18. The main classes could make use of attributes and static methods for properties of the class that do not depend on the specific instance of the object. There are lots of times where “self” is passed into a method and not used. These parts could be refactored to follow better object-oriented coding principles.

Response: Modified the code to use attributes or static methods when better suited.

<Reviewer #3>

Remarks to the Author

Remarks on code availability

Point-by-point response to Comments from the reviewers and Editor:

We thank the reviewers for their careful reading and positive assessment of our revised manuscript. We are pleased that the revisions have clarified the points raised in the initial review. We have addressed the remaining comments and suggestions, particularly those from Reviewer #2, as detailed below. Reviewer comments have been included in their entirety and are in black text. Responses are in blue text.

<Reviewer #1>

Remarks to the Author

The authors have addressed most of my queries. The main problem - that epitope modifications are unlikely to have clinical relevance could as expected not be addressed as this is inherent to the theme of the manuscript.

Nevertheless, the new software may be of value for preclinical experiments.

Response: We thank the reviewer for the thorough evaluation of our manuscript. We acknowledge that the direct clinical translation of our approach is a limitation of this study and that a true evidence of clinical evidence could only occur in side by side clinical testing which is not feasible. Our primary goal was to establish a new methodology and software that could be valuable for preclinical experiments to de-risk AAV gene therapy candidates, a point the reviewer kindly noted. To clarify the scope and limitations of our work for future readers, we have added a statement to the Discussion section of the manuscript.

“Although the specific epitope substitutions evaluated here may not directly translate to clinical settings, this study provides a mechanistic demonstration and methodological foundation for rational immunogenicity reduction.”

<Reviewer #2>

Remarks to the Author

We thank the authors for their point-by-point response to all of our previous comments.

We note that updating the MHC presentation model to the latest generation (i.e. NetMHCIIpan4) leads to prediction of both wildtype and mutant peptides as “presentable”, meaning the same mutants would not have been highlighted for downstream testing. However, we equally appreciate that it is not always the case that more recent deep learning models that have higher headline accuracy achieve this via improved understanding of the underlying mechanisms of the biology, especially in high noise/incomplete data settings.

While we appreciate the inclusion of a discussion point relating to this, I think the Table presented in response to comment #3 should be included in the Supporting Information of the paper to appropriately convey the differences of conclusion reached by different NetMHCII models.

As only one case study is examined here, it is unclear the extent to which we might expect the interpretations offered by NetMHCII2.3 specifically to be the most accurate in other cases. This may affect the perceived impact of the work.

Response: We thank the reviewer for this constructive suggestion regarding the comparison of NetMHCII models. We concur that the comparison table clearly demonstrates the substantial divergence in predictions across the six different algorithms. This finding reinforces our core

argument: that the primary value of the EMMP methodology lies in providing a systematic approach for candidate generation, underscoring the necessity of experimental validation regardless of the specific computational model employed. We have formally included the comparison data as Supplementary Table 2 in the Supporting Information, and the main text has been updated to refer to this table and discuss its implications.

We have added the following text to the manuscript in the appropriate place in the results and the discussion:

“However, we acknowledge a critical limitation: different prediction algorithms can yield substantially conflicting results. As shown in Supplementary Table 2, while NetMHCII2.3 predicted our selected mutations (R312H and R312Q) as having significantly reduced MHC presentability, the more recently recommended NetMHCIIPan4.0 EL algorithm predicts these same mutations would retain strong MHC presentability similar to wild-type. This discrepancy means that using newer algorithms, these mutations would not have been selected for experimental testing, highlighting how algorithm choice fundamentally impacts candidate selection. The current version of EMMP has been updated to use NetMHCIIPan4.1, but this comparison underscores that EMMP's primary value lies in systematic automation rather than enhanced prediction accuracy”

“Importantly, these predictions were generated using NetMHCII2.3. When the same sequences were analyzed using more recent algorithms (NetMHCIIPan4.0 EL), substantially different results were obtained (Supplementary Table 2), demonstrating the algorithm-dependent nature of computational predictions and reinforcing the necessity of experimental validation for any computationally selected candidates”

“This finding reinforces our core argument: the primary value of the EMMP framework is not in the absolute accuracy of any single predictive model, but in providing a systematic, automated pipeline for candidate generation. This divergence underscores the critical importance of experimental validation to bridge the gap between computational prediction and actual biological reality”

-We appreciate the reply and manuscript change implemented in response to our minor comment #8, but request the authors thoroughly review all other parts of the manuscript to ensure they are not claiming that NetMHC can predict immunogenicity when it can only predict peptide presentability. E.g. the textual addition in response to Major comment #3 “While NetMHCII2.3 predicted R312H and R312Q as having reduced immunogenicity” should strictly be “While NetMHCII2.3 predicted R312H and R312Q as having reduced presentability”, etc.

Response: We fully agree that NetMHC is a tool to predict peptide presentability or binding affinity, not immunogenicity. We have thoroughly reviewed the manuscript and corrected all instances where 'immunogenicity' was incorrectly used in conjunction with NetMHC predictions, replacing it with 'presentability' throughout the manuscript (e.g., in the Results and Discussion sections).

-The edits to Figure 6 have improved the figure, but a label should be added to the colour bars for ease of interpretation.

Response: The color bars (legends) representing the different AAV variants were previously included only in the bar chart panels (B, D, and F). To ensure the color-coding is clear and consistent throughout the entire figure, we have now added the corresponding labels (legends) to panels A, C, and E as well.

-We appreciate the authors splitting Figure 2A into two separate figures. Please could they confirm that Dunnett's test was also performed with a non-parametric version of a t-test? Also, now that E and F have been added, would a non-parametric test be more appropriate there as well?

Response: For Figure 2A and 2B, the statistical analysis used the non-parametric Friedman test followed by Dunn's multiple comparisons test (not Dunnett's test, as originally stated). We also corrected a spelling error in the figure legend.

For Figure 2E and 2F, we acknowledge the reviewer's suggestion regarding non-parametric tests. However, given the experimental design where CD8+ T cells showed no detectable response (all values = 0) while CD4+ T cells and total splenocytes showed clear responses, formal statistical testing was not appropriate due to zero variance in the CD8+ group and the small sample size ($n=3$). We have removed the statistical significance indicators (asterisks) from Figure 2E and 2F and updated the figure legend to reflect this limitation. The biological conclusion remains clear: the complete absence of response in CD8+ T cells compared to the consistent responses in CD4+ T cells demonstrates that this epitope is specifically recognized by CD4+ T cells. We have revised the figure legend for 2E and 2F to read: "*Values are presented as mean \pm SD. Statistical analysis was limited by small sample size ($n=3$) and zero variance in the CD8+ group. The biological difference is evident with CD8+ T cells showing no detectable response compared to robust responses in CD4+ T cells.*"

-As a final observation (which we only realised upon re-reviewing the manuscript): it may be appropriate to assess new mutants in all new potential sliding windows of the protein sequence that contain that mutation, to ensure that a new presentable epitope has not inadvertently been created in a different N-mer frame. This check could optionally be introduced as a natural extension of the current EMMP algorithm.

Response: We thank the reviewer for this thoughtful observation. We agree that, after identifying candidate substitutions, it is important to confirm that the mutation does not inadvertently introduce a new presentable epitope in an alternative sliding register. We are working on version 2.0 of the EMMP code based on feedback by internal users and this improvement will be incorporated in the next version of the code.

Remarks on code availability

The authors addressed our comments about the codebase in their point-by-point responses and implemented changes that addressed our previous concerns. Our final point in this review could optionally be implemented as an extension to the current workflow.

<Reviewer #3>

Remarks to the Author

Remarks on code availability
